



# 1 Atmospheric CO$_2$ observations and models suggest strong
# 2 carbon uptake by forests in New Zealand

**K. Steinkamp[1], S.E. Mikaloff Fletcher[1], G. Brailsford[1], D. Smale[1], S. Moore[1],**
**E.D. Keller[2], W.T. Baisden[2], H. Mukai[3] and B.B. Stephens[4]**
[1]{National Institute of Water and Atmospheric Research, Wellington, New Zealand}
[2]{GNS Science, Lower Hutt, New Zealand}
[3]{National Institute for Environmental Studies, Tsukuba, Ibaraki, Japan}
[4]{National Center for Atmospheric Research, Boulder, Colorado, USA}
Correspondence to: K. Steinkamp (kay.steinkamp@gmail.com)
**Abstract**
A regional atmospheric inversion method has been developed to determine the spatial and
temporal distribution of CO$_2$ sinks and sources across New Zealand for 2011-2013. This
approach infers air-sea and air-land CO$_2$ fluxes from measurement records, using back-
trajectory simulations from the Numerical Atmospheric dispersion Modeling Environment
(NAME) Lagrangian dispersion model, driven by meteorology from the New Zealand
Limited Area Model (NZLAM) weather prediction model. The inversion uses *in situ*
measurements from two fixed sites, Baring Head on the southern tip of New Zealand's North
Island (41.408°S, 174.871°E) and Lauder from the central South Island (45.038°S,
169.684°E), and ship board data from monthly cruises between Japan, New Zealand and
Australia. A range of scenarios is used to assess the sensitivity of the inversion method to
underlying assumptions, and to ensure robustness of the results. The results indicate a strong
seasonal cycle in terrestrial land fluxes from the South Island of New Zealand, especially in
western regions covered by indigenous forest, suggesting higher photosynthetic and
respiratory activity than is evident in the current *a priori* land process model. On the annual
scale, the terrestrial biosphere in New Zealand is estimated to be a net CO$_2$ sink, removing 98
(±37) Tg CO$_2$ yr$^{-1}$ from the atmosphere on average during 2011-2013. This sink is much
larger than the reported 27 Tg CO$_2$ yr$^{-1}$ from the national inventory for the same time period.
The difference can be partially reconciled when factors related to forest and agricultural
management and exports, fossil fuel emission estimates, hydrologic fluxes, and soil carbon
change are considered, but some differences are likely to remain.



## 1 Introduction

The exchange of carbon between the atmosphere and the earth's oceans and terrestrial biospheres plays a crucial role in climate projections (IPCC, 2013; Friedlingstein et al., 2014). Predicting the future trajectories of atmospheric $CO_2$, temperature and precipitation requires a solid understanding of how these fluxes are regionally distributed, and how and why they vary on seasonal to decadal timescales. National greenhouse budgets are especially important in light of current policies regarding climate change, such as the annual reporting requirements under the United Nations Framework Convention on Climate Change (UNFCCC).

New Zealand's National Inventory Report (NIR) is compiled by the Ministry for the Environment (MfE), and published annually. For 2013, the NIR puts New Zealand's total $CO_2$ emissions at about 35 Tg $CO_2$ $yr^{-1}$, and it is estimated that the land-use and forestry sector acted as a sink for carbon by removing three quarters of that (27 Tg $CO_2$ $yr^{-1}$) from the atmosphere (MfE, 2015). These estimates are based on measurements from a network of forest plots throughout New Zealand, regularly updated land use maps, and models. Inventory methods give precise estimates of carbon uptake and release of the locally present vegetation type, but are often difficult to scale up to regional or country scales due to heterogeneous biome composition (Ciais et al., 2010). Independent methods are needed to verify the reported carbon sink.

On very large, i.e., continental or global scales, both prognostic land models and inverse atmospheric models have been used (Gurney et al., 2004; Mikaloff Fletcher et al., 2007; Gruber et al., 2009; Le Quéré et al., 2013; Steinkamp and Gruber, 2013; Friedlingstein et al., 2014). Global inversions are a valuable, top-down, tool to estimate large-scale sinks and sources by combining $CO_2$ observations from a global network with atmospheric circulation models. An inverse model interprets the observations to yield an optimized carbon flux distribution that is most consistent with the atmospheric $CO_2$ data. In this approach, atmospheric model simulations relate fluxes at the surface with concentration changes at the observing sites. However, the number of available observing sites and the model resolution are usually insufficient to constrain $CO_2$ exchange on smaller, i.e., regional to country scales.

To address those scales, regional atmospheric $CO_2$ inversions have been developed and used to estimate the carbon budgets of regions like Europe and the USA as well as individual nations (Lin et al., 2003; Stohl et al., 2009; Bergamaschi et al., 2010; Manning et al., 2011). A



regional inversion can provide top-down $CO_2$ exchange estimates from atmospheric $CO_2$
measurements and Lagrangian model simulations that describe the source or sink regions
influencing each measurement. They are complementary to bottom-up inventories and
provide a means to verify national inventories.
Like their global counterparts, regional inversion methods combine $CO_2$ observations from
surface sites with modeled atmospheric circulation to derive the distribution of sinks and
sources over an area of Earth's surface (the inversion domain). In a regional inversion,
however, the sites are not distributed globally and the domain is typically the size of a country
or continent, which poses some additional challenges compared to their global counterparts
(Manning, 2011). For example, an accurate model of background concentrations or sinks and
sources from outside the inversion domain is required, i.e., a baseline. There is also a need for
adequate spatial and temporal resolution for both the estimated fluxes and the circulation
model, to account for topographic effects and local emission gradients and hotspots. Many
regional inversions use continuous *in situ* observations from one to a few measurement
stations to sample the whole domain over the course of days to a few weeks. They use air
from a background-sector to construct a baseline time series (Manning et al., 2011; Uglietti et
al., 2011). Due to chemical inertness of $CO_2$, atmospheric loss processes can be neglected
once the gas has entered the domain, making this approach viable as long as the measurement
station is positioned so that background air can be observed for significant fractions of time. It
is generally of advantage to use multiple stations, which are sensitive to a larger surface area
and allow for a better interpretation of spatial gradients in atmospheric $CO_2$.
In their inversion study, Stohl et al. (2009) estimate emissions for three HFC and HCFC
greenhouse gases on national to global scales for 2005-2007. Their approach uses the
FLEXPART Lagrangian model to describe the recent air history arriving at nine observation
stations distributed globally. They use *a priori* emission maps and estimate both the baseline
and the regional emissions as part of the inverse modeling. Manning et al. (2011) use 20 years
of *in situ* $CH_4$ and $N_2O$ observations from a single station, Mace Head, on the west coast of
Ireland. Mace Head regularly receives air from the midlatitude North Atlantic as well as from
the UK and continental Europe, which allows them to estimate both the baseline and
terrestrial emissions. Their emission estimates for the UK have been used to complement
those reported to the UNFCCC for the period 1990-2007.





Here, we present the first regional inversion for New Zealand, which leverages the country's
unique characteristics. New Zealand is an isolated country surrounded by approximately 2000
km of ocean on all sides. This simplifies the construction of an accurate baseline model, as
$CO_2$ signals from other land masses, especially in Australia and the Northern Hemisphere will
be significantly diluted and become part of the baseline before reaching the country. The
expected national carbon sink, which is estimated by the inversion, is a large fraction of the
fossil fuel emissions, which are prescribed – about three quarters according to the NIR. In
addition, New Zealand has multiple atmospheric $CO_2$ measurement sites across a relatively
small country.
Our inverse model is based on *in situ* observations from two observing stations in New
Zealand, between 2011-2013, and ship board measurements from a regular transect between
Australia, New Zealand and Japan (conducted by Japan's National Institute for Environmental
Studies, NIES). The Numerical Atmospheric dispersion Modeling Environment (NAME III)
Lagrangian model (Jones et al., 2007) was combined with meteorological output from the
New Zealand Limited Area Model (Davies et al., 2005) at ~12 km resolution (NZLAM-12),
to model the pathway of air arriving at the stations. Land model simulations from Biome-
BGC (Thornton et al., 2005) are used as *a priori* estimates. We compare our results to New
Zealand's NIR (MfE, 2015), point out differences and implications, and discuss the regional
distribution and seasonal cycle of $CO_2$ sinks and sources across the country.

## 2   Observations

We use *in situ* measurements of $CO_2$ from two inter-calibrated stations in New Zealand
(**Figure 1**). Baring Head (BHD) is located on the south coast of the North Island, while Lauder
(LAU) is located in the central South Island (**Figure 2**). Both stations are sensitive to different
source regions of $CO_2$ and complement each other to allow for a comprehensive regional
coverage spanning the Southern Ocean, Tasman Sea, the South Island, and – to a lesser
degree – the North Island and the subtropical South Pacific.
The instruments used at LAU and BHD are operated with reference gases traceable to the
World Meteorological Organisations mole fraction scale as maintained by the Central
Calibration Laboratory (CCL) at the U.S. National Oceanic and Atmospheric Administration
(NOAA). Both instruments share common data processing code, improving data inter-
comparability between the sites (Brailsford et al., 2012). In addition, fine scale instrumental





biases were assessed by using a suite of four transfer standard tanks with trace gas
concentrations defined by the CCL. These instrument specific, multiplicative scalings were
applied to the processed hourly data before the inversion. For typical ambient mole fractions
of $CO_2$ (i.e. 380-410 ppm) these adjustments were generally less than 0.07 ppm and 0.1 ppm
for BHD and LAU, respectively.
Some of the elements of this study are prepared to include data from a third station, Rainbow
Mountain (RBM), located in the northern half of the North Island (**Figure 2**). For example, the
region definitions in section 5.2 include a local RBM region. Because the calibration and
quality control processes are equivalent to BHD and LAU, the station can be integrated
seamlessly into the network. It is ideally located to extend the sensitivity of the inverse model
into the north. However, it is not incorporated in this study, because continuous $CO_2$
measurements from RBM were not yet available through 2011-2013.
For this study the hourly mean $CO_2$ records from BHD and LAU covering 1 January 2011 to
31 December 2013 are used. Both stations measure *in situ* with near-continuous observations
throughout the day. The observations are strongly influenced by local signals at night and
under certain meteorological conditions, e.g., a shallow boundary layer (BL) or low wind
speed (Stephens et al., 2013). Measurements at times with deep, well-mixed BL are better
suited for inversion modeling as they are sensitive to sinks and sources from a wider region
and not subject to localized processes or complex atmospheric structure. Similarly, because
the vertical resolution of the meteorological model NZLAM (section 3) is not fine enough to
resolve the exact height of a station inlet, conditions with a well-mixed BL are preferable.
Afternoon observations are, on average, well-suited. Analysis of the diurnal cycle of $CO_2$
concentrations at the various inlet heights of both BHD and LAU shows least variability with
altitude in the 13:00-16:00 afternoon hours. For the inversion, two hourly average data points
per day are selected in the afternoon, 13:00-14:00 local time (LT) and 15:00-16:00 LT. Local
time represents NZST (NZST = UTC + 12 hours) in winter and NZDT (NZDT = UTC + 13
hours) in summer.
For both stations, one standard deviation of 5-minute data about the hourly mean is assumed
as random data uncertainty. This uncertainty is generally much greater than the measurement
imprecision, as it reflects real atmospheric variability, and is instead intended to capture
representativeness errors in both the measurement failing to represent the mean of a model
box and the model failing to represent the specific conditions at an individual location.





## 2.1 Baring Head

BHD station (Lowe et al., 1979) is located at 41.408°S, 174.871°E, 85m AMSL, approximately 10 km southeast of the Wellington urban area (**Figure 2**), close to the edge of a south facing coastal cliff. The surrounding land is sparsely populated and has primarily been used for low density livestock farming. Wind speeds at the site regularly exceed 10 ms$^{-1}$, reducing the influence of local sources. The wind directions are primarily bi-modal with the dominant wind from the north and the secondary direction from the south (Stephens et al., 2013). Southerly air arriving at the site has often been traveling over the Southern Ocean for at least 4 days, sometimes weeks, without any contact to terrestrial sinks or sources of $CO_2$. The station is ideally situated to determine baseline levels of atmospheric $CO_2$ at latitudes up to 70°S during these conditions. At other times, BHD measures air that has recently travelled over Australia or New Zealand, carrying a terrestrial signal of $CO_2$ sinks and sources.

We use hourly averaged $CO_2$ data from the non-dispersive infrared (NDIR) analyser (Ultramat 3, Siemens) *in situ* observations during the 13:00-14:00 and 15:00-16:00 time windows (**Figure 1**) from a 10 m air inlet height. For more details on measurements, calibration and data processing, we refer to Brailsford et al. (2012); Stephens et al. (2013).

## 2.2 Lauder

LAU station is located at 45.038°S, 169.684°E, 370m AMSL, in a broad river valley in the South Island, approximately 35 km north of the township of Alexandra (population 5000). The surrounding land is sparsely populated and largely used for low density livestock farming and seasonal cropping. The local wind direction is predominantly ranging from north-westerly to south-westerly. To the west lies a valley system in mountainous terrain, behind which the north-south running mountain range of the Southern Alps divides the island into a western coastal strip with a humid maritime climate and the eastern part with a more continental climate and relatively clear unclouded skies.

At LAU, $CO_2$ has been measured *in situ* using a dual cell NDIR analyser (LI-7000, LI-COR Inc) since 2008 from a 10 m air inlet height. Unlike the BHD measurements, where detailed descriptions and analyses are given by Brailsford et al. (2012) and Stephens et al. (2013), the LAU measurements have not been published before. Therefore, we provide an extended description of the LAU *in situ* $CO_2$ measurement system in the appendix (Appendix A).





## 3    Model Simulations

We use NAME III (Jones et al., 2007), a Lagrangian dispersion model developed by the UK Met Office driven by three-dimensional meteorological fields precomputed by a numerical weather prediction (NWP) model. While initially developed more than two decades ago as an emergency response particle tool for nuclear outfall, NAME has since evolved into a general purpose dispersion model that is being used from local scales (a few hundred metres) to mesoscales and global scales. Atmospheric turbulence is simulated using a random walk technique (Morrison and Webster, 2005). $CO_2$ is modelled as an inert gas due to its long lifetime in the atmosphere far exceeding the 4 day periods used for the back-trajectories.

In this work, meteorology from the NZLAM-12 model was used to drive NAME. NZLAM-12 is a local configuration of the UK Met Office Unified Model (Davies et al., 2005) with a horizontal resolution of ~12 km and 70 vertical levels up to a ceiling height of 80 km. The meteorology is a sequence of short, 6 hour forecasts with hourly output that are produced from successive NZLAM simulations and cover the period 2011-2013. At the beginning of each simulation the available meteorological observations are assimilated into the model to match the state of the NZLAM atmosphere to the measured atmosphere. NAME uses the boundary layer depth (BLD) from NZLAM and applies a minimum and maximum BLD of 50 m and 4000 m, respectively. The maximum height in NAME is 30 km, corresponding to the first 59 levels of NZLAM. Both NZLAM and NAME cover a domain ranging from 146.8 E to 185.8 E in longitude and from 53.4 S to 26.0 S in latitude (inversion domain in **Figure 5**).

### 3.1    Air history and station footprints

The NAME model is run in backward mode to analyse the history of the air traveling towards BHD and LAU over the preceding 4 days. Model particles are released from both stations during a period of 1 hour, twice per day in 2011-2013, at 13:00-14:00 and 15:00-16:00 LT. A simulation period of 4 days was found sufficiently long to allow all particles to leave the domain during most meteorological conditions, except during extended periods of very low windspeed. An air history map has been calculated for each release (**Figure 2**). We use model output that represents the 4-day integrated air concentration (also called dosage, unit g s m$^{-3}$) inside each grid box on a regular 0.1°x 0.1° grid, designed to be very similar to the ~12 km grid of NZLAM-12. During each release, the dispersion of 10,000 particles is modelled and every particle registered within the boundary layer at a given time contributes to the dosage of the respective grid cell. Particles are simulated in 3 dimensions and do not disappear when





leaving the boundary layer as long as they remain below the maximum model height of 30
km. Particles can leave the boundary layer temporarily and descend back into it at a later time,
in which case they would again contribute to the dosage. An example of this can be seen in
**Figure 2**, where many particles leave the boundary layer just south of the South Island and
later (from the point of view of the backward simulation) descend again, visible as a weaker
dosage (less strong colours) for some stretch of the map. A dosage map for a station is also
called that station's footprint.
Average footprints for the BHD and LAU stations were computed by summing the footprints
for every day and release period in 2011-2013 and normalizing them such that the domain
integral equals one (**Figure 7**). These footprints represent the average sensitivity of a station to
spatially distributed surface fluxes (sinks and sources) of $CO_2$. They have also been used to
help inform the partitioning of the inversion domain into a set of regions for which weekly
surface fluxes are calculated (Section 5.2).
### 3.2  Transport matrix
For particle transport, the mass flux during each 1 h release period is 1 g $CO_2$ s$^{-1}$, amounting
to a total emission of 3600 g $CO_2$ over the period (0.36 g $CO_2$ is assigned to each particle).
The flux strength is an arbitrary choice and does not affect the transport results due to the
implied linearity of transport. A transport matrix T (unit s m$^{-1}$) is formed by dividing the
dosage by the total emitted mass and multiplying by the area (m$^2$) of each surface grid cell.
Each element of T describes the atmospheric transport of a continuous emission of 1 g $CO_2$
m$^{-2}$ s$^{-1}$ from a given grid cell over the previous 4 days and subsequent contribution to the air
concentration at the receptor (BHD or LAU) during each 1 h period. With **x** being a vector
containing all grid cells and **c** a vector containing the concentration (unit g $CO_2$ m$^{-3}$) for all 1
h periods, this is written as
$$T\boldsymbol{x} = \boldsymbol{c} \tag{1}$$
Given T and the measured concentrations **c**, the inversion developed in this work solves for
the $CO_2$ fluxes **x** using a Bayesian optimisation, i.e., a statistical model that balances
information from measurements with *a priori* knowledge about the fluxes (section 6). Instead
of solving on the grid scale, the fluxes in **x** are pre-aggregated into a set of regions and *a*
*priori* flux maps are taken into account for the terrestrial and oceanic portions of the domain
(section 4).





**4    *A Priori* CO$_2$ Flux Maps**
The Bayesian approach in this study uses spatially distributed information about CO$_2$ sinks
and sources as first-guess, or *a priori*, fluxes for terrestrial and oceanic regions. These fluxes
are optimized by the inversion using the constraints imposed by the CO$_2$ measurements at the
stations (section 6). Fossil emissions are accounted for as well, though unlike the natural
fluxes they are prescribed and not optimized by the inversion. Here we describe the data sets
and flux maps used, while their incorporation in the inversion is described in section 5.
**4.1    Terrestrial**
First-guess land-to-air CO$_2$ fluxes from the biosphere for every month in 2011-2013 are
obtained from the Biome-BGC model (Thornton et al., 2005). Biome-BGC (v4.2 final
release) is an ecosystem process model that estimates the storage and flux of carbon, nitrogen
and water (Thornton et al., 2002). The model has been extensively tested and validated for
North American and European ecosystems, and in addition was recently extended and applied
to New Zealand managed pasture systems (Keller et al., 2014). The adaptation of the Biome-
BGC model to New Zealand by Keller et al. (2014) is used in this study to estimate net
ecosystem production (NEP) for 5 biomes across New Zealand: dairy pasture, sheep and beef
pasture, shrub, evergreen broadleaf forest (EBF), and evergreen needleleaf forest (ENF). The
model is driven by daily weather data from the NIWA virtual climate station network
(VCSN). VCSN data include numerous meteorological parameters on a regular (~5 km) grid
covering the whole of New Zealand (Tait et al., 2006). The data are based on the spatial
interpolation of actual data observations made at climate stations located around the country.
Soil attributes are incorporated from the Fundamental Soil Layers database (Landcare, 2015).
Biome-BGC produces NEP maps for each biome covering the whole country, i.e., it does not
make assumptions about the actual distribution of biomes. In order to partition the country
into biomes approximating the five categories available in Biome-BGC and then mask and
sum the NEP contributions from each biome, we produced a land-cover/land-use (LCLU)
map. The LCLU map uses 10 categories based on a combination of the land cover database
(LCDB) for New Zealand (Shepherd and Newsome, 2009; Dymond et al., 2012) and the
Land-Use in Rural New Zealand (LURNZ) model (Hendy et al., 2007; Timar, 2011; Kerr et
al., 2012).
The New Zealand LCDB is a thematic classification of land-cover and land-use categories,
created using satellite imagery and covering all of mainland New Zealand. Version 3 was



used here, which contains 33 categories for each of three periods; summer 1996/97, summer
2001/02, and summer 2008/09. The dataset is polygon-based and designed to be compatible
in scale and accuracy with Land Information New Zealand's 1:50,000 topographic database.
For the purpose of this study the distribution of land-cover types for 2008/09 were used and
rasterized on a 5 km x 5 km grid.
LURNZ is a dynamic partial equilibrium model that simulates changes in private rural land
use over time and space. It focuses on four key land uses – dairy, sheep and beef, forestry
(plantations), and scrub/shrubland. While the model's primary focus is on simulating future
changes in land-use under scenario projections of commodity prices in one of the four sectors,
it also provides a baseline of actual land-use in 2008. This 2008 basemap is used in this study
to match the Biome-BGC dairy and sheep/beef pasture biomes; however, LURNZ does not
include native forests.
To account for all biomes in Biome-BGC the LURNZ 2008 basemap and the LCDB 2008/9
land-cover map are combined as follows. First the 33 LCDB categories are aggregated into 7
– forest, scrub and shrubland, grassland, cropland, water bodies, bare or lightly-vegetated
surfaces, and artificial surfaces. The forest and grassland categories are then sub-divided into
plantations, "other forests", dairy pasture, sheep and beef pasture, and "other grasslands"
using LURNZ, which results in the LCLU map in **Figure 3**a.
The ENF biome is assumed to be well represented by the plantation forest category, with
plantations consisting primarily of pine trees. The EBF biome is assumed to be better
represented by the "other forests" category. The categories of artificial surfaces, bare/lightly-
vegetated surfaces, and water bodies are assigned a zero flux, i.e., no exchange of $CO_2$ with
the atmosphere. No flux estimates are made for cropland and "other grasslands"; this does not
affect results significantly, because these categories represent only a small portion of the total
land area. **Figure 3**b shows the 2011-2013 mean *a priori* land-to-air $CO_2$ flux as estimated by
matching the LCLU and Biome-BGC biomes in this manner and summing their contributions
to the overall NEP. The monthly and annual contributions are shown in **Figure 3**c. Weekly
first-guess $CO_2$ flux maps are obtained by simple interpolation of the monthly estimates
throughout 2011-2013.
An uncertainty estimate is computed for the *a priori* $CO_2$ flux from each grid cell. Based on
Keller et al. (2014) and personal communication with the authors, we assign a 10%
uncertainty for pasture land. For forests, we assign 10% everywhere except in the Canterbury



and Otago regions in the South Island, where 56% and 36% are used, respectively. These are
conservative estimates based on a comparison of the Biome-BGC modelled live stem carbon
with the national exotic forest regional yield tables (MPI, 2012). The Canterbury and Otago
regions were assigned larger uncertainties to reflect the larger discrepancy between the
Biome-BGC model and the yields in these regions. The uncertainty is taken into account by
the Bayesian optimization (Section 5).
**4.2   Oceanic**
First-guess air-sea $CO_2$ fluxes are calculated based on a global dataset of surface ocean $pCO_2$
(Takahashi et al., 2009a). The dataset contains approximately 4.5 million measurements of
surface water partial pressure of $CO_2$ ($pCO_2$) obtained over the global oceans during 1968-
2008, approximately 90,000 of which were taken inside the model domain of this study. A
monthly climatology on a global 4x5 grid was derived by Takahashi et al. (2009b), which also
includes an estimate of air-sea $CO_2$ flux derived from the difference of surface ocean and
atmospheric $CO_2$ and a gas exchange rate following Wanninkhof (1992).
An uncertainty estimate for the *a priori* ocean fluxes is computed as the root mean square of
two components reflecting the uncertain gas exchange rate and the spatiotemporal coverage of
measurements inside grid cells. For the first component we recalculated the $CO_2$ flux using
each of 7 additional gas transfer models (Ho et al., 2006; Sweeney et al., 2007) and used one
standard deviation from the 8-model mean as uncertainty. The second component applies an
uncertainty to grid scale fluxes inversely proportional to the number of measurements taken
inside them for a given month of the climatology (Steinkamp and Gruber, 2013). As with the
terrestrial $CO_2$ flux prior, the uncertainty is accounted for in the Bayesian inversion.
**4.3   Fossil emissions**
A gridded map of $CO_2$ emissions is derived from the Emission Database for Global
Atmospheric Research (EDGAR) version 4.2 (JRC, 2011). EDGAR contains global emission
inventories for greenhouse gases and air pollutants from sectors including energy, industrial
processes, solvents and other product use, agriculture, land-use change and forestry, and
waste. Annual emissions are available on a 0.1°x0.1° grid over the globe up to the year 2010.
Emissions for the 2011-2013 time period were approximated by extrapolation using the trend
in global total emissions over 2000-2010. The spatial distribution was assumed unchanged
from 2010 (**Figure 4**). Total emissions for the New Zealand mainland are 47.8 Tg $CO_2$ yr$^{-1}$ in

32  2011-2013.





Fossil $CO_2$ emissions are not optimized by the inversion, but their contribution to the $CO_2$
signal at both stations (**Figure 8**) is subtracted from the actual measurements beforehand. That
contribution is calculated using the transport matrix from NAME, i.e., applying Equation (1)
with **x** containing the emissions from every grid cell and week in 2011-13. The vector **c** then
contains $CO_2$ concentrations for the twice-daily release periods at both stations that are caused
by the emissions. To convert concentrations into mole fractions (ppm) the atmospheric
pressure and temperature from the NAME model are used, which were interpolated to the
BHD and LAU site coordinates from the NZLAM-12 temperature and pressure fields.
## 5   Regional Flux Estimation
### 5.1   Baseline analysis
Any regional $CO_2$ inversion can only estimate sinks and sources within the boundaries of the
model domain. Sink and source processes from outside the domain become part of the $CO_2$
background concentrations (i.e., baseline) seen by the regional inversion at the boundary.
Therefore, an accurate description of this baseline is needed. A common approach is the
background-sector method (Manning et al., 2011; Uglietti et al., 2011), where air is classified
as baseline if it originates from a certain wind sector and fulfils site specific meteorological
criteria. A continuous baseline is constructed using gap-filling, which is subtracted from all
other measurements before the inversion. The inverse model then interprets these differences,
or anomalies, to find the optimal distribution of sinks and sources within the model domain.
The background-sector method has been applied to the BHD $CO_2$ record by Brailsford et al.
(2012) and Stephens et al. (2013). They use steady background $CO_2$ mole fractions during
southerly wind conditions at BHD and apply a multi-step filter to the BHD record to obtain a
$CO_2$ baseline representative of a large region over the Southern Ocean. In short, the filter
selects measurements during extended periods of southerly winds at the site, during which a
maximum standard deviation of 0.1 ppm is achieved. Additional meteorological conditions
must be fulfilled to preclude the influence of local sources and to ensure the air has not passed
over the South Island before arriving at BHD. After filtering the data for baseline conditions,
a continuous baseline is constructed using the seasonal time series decomposition by Loess
(STL) algorithm (Cleveland et al., 1990), which can be sampled hourly, i.e., during the 13:00-
14:00 and 15:00-16:00 LT release periods. The baseline derived from the BHD record is
shown in red in **Figure 1** and will be called the southern baseline.



One disadvantage of a background-sector approach based on a single site is that it may not
capture variability in background concentrations from different wind conditions, in particular
along the latitudinal axis with its gradient in atmospheric $CO_2$, which could lead to biases in
the flux estimates within the domain. In our case, BHD's background sector is ideally situated
to obtain a $CO_2$ baseline representative of a large region over the Southern Ocean. However,
observations made during northerly events are not always well described using this baseline,
as the air often originates from the northern Tasman Sea or the subtropical South Pacific and
carries a contribution from Northern Hemisphere $CO_2$ (Section 6, **Figure 7**).
To alleviate this we augment the southern baseline with a second baseline from ship data
representative of the northern sector. This northern baseline is based on *in situ* $CO_2$
observations using a NDIR analyser on board the Trans Future 5 (TF5), a ship of opportunity
that cruised the triangle Japan/Australia/New Zealand about once a month during the period
2011-2013 (Chierici et al., 2006). We mask out data points from along the ship track (**Figure 5**)
to keep observations from the open ocean and avoid observations taken close to the land,
especially near the Australian east coast as it is located upwind during average south-westerly
conditions and hosts large urban centres with significant $CO_2$ emissions. These data are then
latitudinally averaged between 26-27°S to produce a baseline representative of the northern
edge (26°S) of the inversion domain. A continuous baseline is constructed using the same
STL routine as for the southern baseline.
For both baselines, uncertainty estimates are formed based on the monthly standard deviations
of the *in situ* data as well as differences between measurements and the STL smoothed curve.
A more detailed description of the construction of both baselines is provided in the appendix
(Appendix B).
A combined $CO_2$ baseline is constructed that takes into account where the modelled
trajectories originated for any given data point. The daily NAME station footprints for the
13:00-14:00 and 15:00-16:00 LT windows are integrated along the southern and northern
edges of the domain to determine the relative fraction of back-trajectories leaving the domain
to the south and north. These fractions are then used to weigh the two baselines and create a
baseline associated with each of the twice-daily data points. Uncertainties are weighed in the
same way. The combined baseline is shown in green in **Figure 1**. For the plot the 13:00-14:00
and 15:00-16:00 LT weighted baselines were averaged, as they are visually almost
indistinguishable, but the individual baselines are used in the inversion.





**5.2   Regional partitioning**
$CO_2$ fluxes are estimated for every week in 2011-2013 and for 25 geographic regions
distributed across the inversion domain (**Figure 6**). The within-region pattern of the fluxes is
prescribed using the *a priori* flux maps for New Zealand and the surrounding oceans, while
the inversion estimates regional totals. The definition of the regional boundaries was guided
by several factors, including the distance from the measurement stations, the gradient of the
station footprint, local orography, and fossil emission hotspots. For land regions in New
Zealand, additional factors include land-cover and land-use types as well as the expected
(first-guess) $CO_2$ flux distribution.
Due to its large distance from the stations, the portion of Australian land inside the inversion
domain is represented by a single region (#16). No *a priori* information about natural $CO_2$
fluxes from Australia is assumed, i.e., they are set to zero with a very large uncertainty of
1000 Tg $CO_2$ $yr^{-1}$, so that the inversion is free to adjust them. This is based on an analysis
showing generally low sensitivity of $CO_2$ measurements at our stations in New Zealand to
Australian fluxes (Section 6, **Figure 8**). The analysis uses fossil emissions from the Australian
region (section 4.3) and investigates whether these emissions leave a significant imprint on
measured $CO_2$ at BHD and LAU. Except for a few days this imprint is negligible (section 6).
The portions of the Southern Ocean, South Pacific and Tasman Sea that are inside the model
domain were divided into 6 open ocean and 3 coastal regions (#17-25). The open ocean
regions are large to make their regionally integrated contribution to the $CO_2$ signal become
discernible at the stations, though still much smaller compared to the land regions in New
Zealand. They divide the domain into three northern and three southern regions to account for
the difference in ocean biogeochemistry between Southern Ocean and subtropical Pacific
waters, with guidance from patterns of surface ocean $pCO_2$ from the *a priori* map. The coastal
ocean regions were included to separate the open ocean from the land explicitly, and to
account for their stronger influence on the measured $CO_2$ due to their relative proximity to the
stations compared to the open ocean. The coastal ocean was defined as the union of a 60 km
coastal band around New Zealand and the portion of ocean with a mean 2011-2013 BHD
footprint value above a fixed threshold. The threshold was chosen such that the integrated
$CO_2$ signal at BHD from coastal regions is 25% of that from all ocean areas.
It is generally important to separate regions that exhibit a strong variability in sensitivity, as
otherwise these within-region gradients can skew the regional totals estimated by the





inversion towards the most sensitive areas inside the region. For land regions in New Zealand,
we used the spatial gradients of the 2011-2013 footprints as an estimate for this variability,
similar to the coastal regions, except that for the land, we use the combined footprint of BHD
and LAU (**Figure 9**) and also account for additional factors.
New Zealand was divided into 15 land regions (#1-15) as follows. Three small regions around
BHD, LAU and RBM were defined, which have the largest contributions to the $CO_2$ signal at
the respective stations. This separation of the highly influential local regions from the rest of
the country follows the same rationale as the separation of the coastal from the open ocean,
with the aim to prevent the inversion from allocating local signals to regions further upwind.
A separate region around Auckland and Hamilton was defined to capture the strong fossil
emissions there. The remaining regions were defined with the aim to minimize both the
footprint variability and the expected flux variability inside each region, while accounting for
topographic features and avoiding many different types of land-cover/land-use in the same
region.
The resulting regional partitioning is shown in **Figure 6**. A major feature is the role of the
Southern Alps as a dividing range between the humid west coast of the South island
containing large patches of native forest, and the dryer regions in the central and eastern parts,
where pasture land is predominant. On the North Island, the axial mountain ranges divide the
land into east and west as well, but the distribution of forests and pasture is more complex.
BHD and LAU have relatively low sensitivity to the northern half of the North Island, which
results in large uncertainties after the inversion for individual regions. However, regionally
aggregated results are well constrained in that part of the country.

### 5.3   Inversion methodology

The aim of the inverse method is to estimate a $CO_2$ flux from every region and for every week
between 2011 and 2013 using a Bayesian approach (Gurney et al., 2004; Tarantola, 2005;
Steinkamp and Gruber, 2013). The approach assimilates information from the twice-daily
observations from both stations (the "data") and accounts for *a priori* flux distributions (the
"prior") and contributions from fossil emissions.
The data time series is constructed by subtracting the baseline from the station measurements.
The modelled $CO_2$ signal from fossil emissions is also subtracted. The resulting time series
represents the part of the observed $CO_2$ signal that cannot be explained by background
concentrations or fossil emissions, but is due to the net effect of sinks and sources of $CO_2$



over the ocean and land portions inside the model domain. The data for every 1 h period and
both stations is written as a vector **d**.
Data uncertainty is calculated as the quadrature sum of the baseline uncertainty (Section 5.1)
and the $CO_2$ data uncertainty (Section 2). An additional uncertainty component of 0.4 ppm is
assumed to account for uncertainties in the inverse modeling system as well as possible errors
in the fossil fuel emission estimates. That value is based on a goodness of fit analysis of the
inverse model (reduced chi-squared statistic, as described below). The final data uncertainty is
taken as the root mean square (quadrature) of both components. The square of the uncertainty
populates the main diagonal of the data covariance matrix $C_d$. We assume no correlations
between pairs of data points, so all off-diagonal elements of $C_d$ are set to zero.
The regional prior (denoted $x_0$) is obtained by integrating the weekly *a priori* terrestrial and
oceanic flux maps over each of the 24 non-Australian regions. The prior uncertainty is
similarly obtained by aggregating the grid-scale uncertainty estimates. Since the within-region
flux patterns remain fixed, we assume full spatial correlation when propagating grid-scale
uncertainties to the regional scale. For land regions we added (via root mean square) an
additional uncertainty component of 50% of the seasonal flux amplitude. This is to allow the
inversion to shift the seasonal cycle more freely; without it the seasonal turning points – i.e.
the switch between net $CO_2$ uptake in the summer months and net release in the winter –
would essentially be fixed as the flux is near zero and the grid-scale uncertainty estimates for
the Biome-BGC model are proportional to the flux strength. The diagonal prior covariance
matrix $C_0$ contains the regional uncertainty.
The regional prior is linked to the data vector as in Equation (1), except **x** now contains fluxes
on the regional instead of grid scale, and the transport matrix T links regional total fluxes to
the data time series with baseline and fossil signal subtracted.
The inversion process seeks an optimal solution to the transport equation by balancing the
data and prior constraints (Tarantola, 2005), i.e., by minimizing a Bayesian cost function J
with respect to **x**,

$$J = \frac{1}{2}(Tx - d)^T C_d^{-1}(Tx - d) + \frac{1}{2}(x - x_0)^T C_0^{-1}(x - x_0) + \frac{1}{2}(Sx)^T C_s^{-1}(Sx)$$

$$= \frac{1}{2}\left(\tilde{T}x - \tilde{d}\right)^T \tilde{C}_d^{-1}\left(\tilde{T}x - \tilde{d}\right) + \frac{1}{2}(x - x_0)^T C_0^{-1}(x - x_0) \qquad (2)$$



The first term in the equation evaluates the deviation of the modelled time series from the
data, with each data point weighted with the inverse uncertainty. The second term evaluates
the deviation of the optimized regional fluxes to the prior fluxes. The last term is a Gaussian
smoother being used to limit changes in week-to-week fluxes. The operator S forms a vector
whose elements correspond to the difference of each flux in **x** and the flux of the following
week. The diagonal matrix $C_s$ contains values representing the strength of the smoother. We
chose 5 kg $CO_2$ m$^{-2}$ yr$^{-1}$ for every grid cell, translating into slightly different values on the
regional scale due to varying surface areas. This value is more than ten times larger than the
largest flux from any grid cell of the *a priori* flux maps (**Figure 3**), hence the smoother is very
weak. In fact the smoother was designed to have a negligible effect on estimated $CO_2$ fluxes
and not interfere with the prior and data constraints. Its role is merely to favour solutions with
small week-to-week changes in cases where a second solution with much larger week-to-
week changes would result in a very similar cost, J. Due to their mathematical forms being
equivalent, the smoothing term can be absorbed in the data term in Equation (2) by appending
S to T (forming $\tilde{T}$), $C_s$ to $C_d$ (forming $\tilde{C}_d$), and a zero vector of appropriate length to **d**
(forming $\tilde{\boldsymbol{d}}$). The reduced chi-squared statistic $\chi^2 = 2J/n$ is used to assess the fit of the
inverse model to the observations (Gurney et al., 2004; Baker et al., 2006). The number of
degrees of freedom, i.e. the number of observations minus the number of sources, is denoted
by $n$. Inclusion of the aforementioned data uncertainty component of 0.4 ppm ensures $\chi^2 \approx 1$,
which means that the extent of the match between observations and the model as well as
between the *a priori* and *a posteriori* sources are in accord with their respective uncertainties.
The cost function in Equation (2) is minimized analytically (Enting, 2002; Tarantola, 2005),
to yield *a posteriori* fluxes **x** and covariance matrix C,

$$\boldsymbol{x} = C(\tilde{T}^T \tilde{C}_d^{-1} \tilde{\boldsymbol{d}} + C_0^{-1} \boldsymbol{x_0})$$
$$C = (\tilde{T}^T \tilde{C}_d^{-1} \tilde{T} + C_0^{-1})^{-1} \tag{3}$$

The square root of the diagonal elements of C are reported as uncertainty estimates for the *a*
*posteriori* fluxes.
**5.4   Sensitivity scenarios**
Considerable effort has been undertaken to ensure, e.g., the high quality of available
observations, the inter-comparability of measurements from BHD and LAU, and the use of a



state-of-the-art land process model to provide meaningful first-guess estimates. However,
there are a number of potential sources for bias that cannot be accounted for explicitly, but
could have a significant influence on estimated land fluxes. These include (i) the $CO_2$
baseline, (ii) the modeling in NAME, and (iii) the ocean prior fluxes.
Sensitivity scenarios were designed to address each of these potential biases, as described
below. The results are discussed in section 7.4.
(i) The inverse method assumes that air entering the domain is accurately characterized by the
baseline $CO_2$ time series. While random noise in the baseline concentration is accounted for
(Section 5.1), there remains the possibility of systematic bias. A positive (negative) bias in the
baseline would cause the inversion to estimate a total $CO_2$ flux that is depressed (elevated), in
order to explain the measurements at the stations. It is assumed that this effect is most
pronounced at the edge of the domain, i.e., in the Australian and open ocean regions. To
address its significance to the inner regions, sensitivity runs were conducted with both
positive and negative biases. The baseline mixing ratio is first decreased, then increased, by
one standard deviation, i.e., its uncertainty.
(ii) The $CO_2$ fluxes are assumed constant in time over a one week period and their geographic
distribution within each region is fixed. A $CO_2$ flux pulse lasting only a few hours cannot be
resolved and could bias the weekly average flux if it coincides with a high sensitivity during
the pulse. Otherwise, its contribution will be correctly contained in the weekly average flux,
unless the region is being unevenly sampled. That is, if a specific observation is sensitive to
only a small area inside the region, then the flux estimate for the entire region will be biased
towards that area, which may not be representative for the region. This is why we took the
geographic distribution of biomes into account when defining the regions. The number of
different biomes was minimized and isolated patches of biomes avoided inside each region.
However, the region definition remained subjective, so we included a sensitivity case where
the within-region flux pattern is flat, i.e., the flux is constant region-wide. Not all potential
biases are removed this way, as that would require solving the inverse problem at a much
higher resolution, but it gives an indication of the influence of a particular choice of pattern.
(iii) Estimates of terrestrial $CO_2$ fluxes in New Zealand are influenced by the ocean flux prior
through atmospheric transport. After entering the model domain at baseline levels, the air
travels inevitably over a large stretch of ocean and will arrive at the New Zealand coast
carrying an oceanic signal in its $CO_2$ concentration and the difference to the measurements at





the stations will be interpreted by the inversion in terms of terrestrial $CO_2$ flux. In a sensitivity
test, we excluded the ocean prior to isolate its impact on the results.

### 6   Analysis of New Zealand's *in situ* $CO_2$ Observing Sites

We conducted a clustering analysis using NAME III to characterise the catchment areas of the
BHD and LAU stations. The clustering was performed using a convergent k-means
procedure, which is based on Kidson (1994), but adjusted slightly to allow a larger number of
trajectories to be clustered, i.e., by using a smaller number of random seeds. This significantly
boosts the computation at the expense of likelihood to find the global minimum, however, the
reduced number of seeds appeared large enough to come sufficiently close, as repeated
computations with randomly different subsets all produced very similar results.
A set of 1000 trajectories was used between 15:00-16:00 LT for every day in 2011-2013,
resulting in approximately 1 million trajectories for each station. The number of clusters was
set to 7, because this number maximised the distinctness of clusters with respect to each other
as obtained from their silhouette values. Cluster centroids and sizes are overlain on the station
footprints in **Figure 7**, together with the geographical width of the clusters.
In addition to the clustering analysis, we applied Equation (1) to the *a priori* flux maps for
every day in 2011-2013. This allows us to calculate the imprint of Australian and New
Zealand fossil emissions as well as oceanic and New Zealand terrestrial sinks and sources on
the $CO_2$ concentration measured at BHD and LAU (**Figure 8**).
$CO_2$ measurements at BHD are most sensitive to sinks and sources in the Southern Ocean
(south of 55°S), the Tasman Sea and the South Island. Australia and the North Island
influence BHD $CO_2$ to a lesser extent. Observations at LAU are strongly influenced by local
to regional terrestrial sinks and sources of $CO_2$, enabling the station to see air from a large
portion of the southern South Island.
The low sensitivity to Australia means it is infeasible to infer Australian $CO_2$ fluxes with our
observational network most of the time. On the other hand, this underscores the isolation of
New Zealand, where air is received that largely contains background concentrations from the
vast body of surrounding ocean. This allows us to estimate terrestrial fluxes in New Zealand
with high sensitivity and little disturbing influence from continental sources.





## 6.1 Baring Head

In 2011-2013, BHD sampled air that has travelled from the Southern Ocean 41% of the time along two cluster pathways (**Figure 7**), which correspond to southerly wind conditions at the site. The more southerly cluster of the two (16%) contains trajectories that mostly have not seen land over at least 4 days and will carry Southern Ocean baseline $CO_2$. Trajectories in the more westerly cluster (25%) have travelled across most of the South Island after originating in the Southern Ocean and will carry a signal of the terrestrial sinks and sources of $CO_2$ there. Another 17% of trajectories are originating from the south-west, but are associated with slower wind speeds, so that within the 4-day timeframe of the back-trajectories they have not yet left the domain. They correspond to a local northerly wind at BHD associated with a common synoptic pattern involving an anticyclone over the Tasman Sea. The Southern Alps on the South Island strongly influence the south-westerly air flow and deflect it northward along the west coast and then through Cook Strait, where it is channelled into a northerly flow by local topography. Trajectories arriving from Australia and the Tasman Sea occur 13% and 9% of the time, respectively. 10% of the trajectories have crossed large parts of North Island before arriving at the station.

The application of Equation (1) to the *a priori* flux maps shows that there are only 4 days in 2011-2013 when a discernible (larger than 0.1 ppm) signal from Australian fossil fuel emissions within the inversion domain was received at BHD. The signal was always smaller than 0.4 ppm, the minimum overall uncertainty assumed in the modeling system.

During the winter and summer seasons New Zealand land is the main contributor to the BHD data series, with a seasonal pattern matching the respiration and growing cycles. Assumed aseasonal fossil emissions from New Zealand (mostly from the nearby city of Wellington) as well as seasonal oceanic fluxes also play an important role at BHD.

## 6.2 Lauder

For LAU, there are two southwest clusters representing a combined 40% percent of trajectories. These are very similar in size to the corresponding clusters for BHD, and have identical source areas in the Southern Ocean. Both clusters differ in whether the air flow leads to local winds at LAU from the west or south. While similar to BHD's southern cluster, the air from the southern cluster would have travelled over a considerable stretch of land before arriving at LAU. 14% of the time, the air being sampled belongs to another southwestern cluster, which originates in the Tasman Sea. In addition, there is a western cluster containing


15% of trajectories that has crossed South Australia and the Tasman Sea as well as two
northern clusters representing air with mixed origin from the northern Tasman Sea or the
North Island. About 14% of trajectories are contained in a slow cluster whose origin is not
very far from LAU. These cases correspond to slow winds at the site and indicate that the
measurements are highly impacted by local and regional sources as the air has been travelling
over nearby land for the preceding 4 days.
The application of Equation (1) to the *a priori* flux maps shows that LAU station is dominated
by terrestrial fluxes from New Zealand (particularly from South Island), with only minor
contributions from the ocean, reflecting its location further inland and shielded from the
predominant westerly winds by the Southern Alps. The seasonal amplitude in the $CO_2$ signal
at LAU is about twice as large as at BHD, due to the more continental climate and more
pronounced growing seasons in central South Island. Similar to BHD, there are only 5 days in
2011-2013 when a larger than 0.1 ppm signal from Australian fossil fuel emissions within the
inversion domain was received at LAU.

## 7   Flux Results and Discussion

### 7.1   Seasonal cycle

The inversion finds a much stronger seasonal cycle than the Biome-BGC model simulations
used as a prior (**Figure 3c**), especially associated with enhanced $CO_2$ uptake during the
growing season in (austral) summer (**Figure 10**). There is very good agreement in the phasing
of the seasons with the land process model during all 3 years, which is particularly
encouraging in light of the weak constraints on the phasing applied through the prior (section
5.3). This strong seasonality is robust within the estimated *a posteriori* uncertainty range and
across the sensitivity cases. Uncertainties for weekly fluxes were reduced significantly
compared to the prior, even when the range of sensitivity cases is added as extra uncertainty.
The enhanced seasonal amplitude is assigned to the South Island almost exclusively, with
much stronger uptake during the growing season compared with carbon uptake in Biome-
BGC. The uncertainties associated with South Island fluxes are generally smaller than on the
country scale, because of the high sensitivity of the LAU and BHD stations to fluxes from
much of the South Island. On the other hand, the North Island is estimated to have a weaker
seasonal cycle, in good agreement with the prior, which can be attributed to widespread areas
of summer soil water deficits, and the more marine climate there, i.e., weaker seasonal



temperature variations and milder winters. Uncertainties for North Island fluxes, especially
from the northern half of the North Island, are generally larger due to the lower sensitivity of
the stations to that area. While northerly breezes are very common at BHD (**Figure 7**), they
often correspond to a situation where southwesterly air was deflected by the Southern Alps
and channelled by local topography to turn into a northerly at the station. Air that has
travelled across the North Island and picked up its terrestrial $CO_2$ signal is therefore less often
sampled at BHD than local wind direction would suggest. At LAU, North Island air can be
sampled only about 8% of the time, based on the NAME cluster analysis. In a future study,
the sensitivity to North Island fluxes can be greatly enhanced by $CO_2$ observations at the
recently established RBM station in central North Island.
When separating the South Island into parts east and west of the Southern Alps, it becomes
apparent that most of the enhanced seasonal cycle occurs, in fact, in the west, despite the
slightly smaller surface area (86,173 $km^2$ compared to 88,348 $km^2$). Along the west coast, the
inversion estimates the seasonal amplitude to be more than twice as large as suggested by the
prior. Tracing the cause further to the individual regions reveals that Fiordland (region #13) is
the strongest contributor to the signal. Fiordland is extremely sparsely populated and covered
to a large extent by indigenous temperate rainforest with southern beeches, fern trees and
shrub. When forming the prior flux map, these forests were categorized as evergreen
broadleaf forest (EBF) and the respective module from Biome-BGC used. However, the EBF
module had not been optimized for New Zealand forests, so it is possible that the Fiordland
forests are not well described by that category. The inversion suggests much stronger
photosynthetic and respiratory activity in these forests than the prior model.

### 7.2   Response to the 2012/2013 drought

The austral summer of 2013 was characterised by unusually high temperatures and low
precipitation over much of New Zealand (Turner, 2013; Blunden and Arndt, 2014), with
sustained periods of severe drought in February-March 2013. The North Island and the west
of the South Island were the most strongly affected regions (Porteous and Mullan, 2013). The
Biome-BGC model is driven by detailed, reanalyzed weather data and clearly shows a
positive flux anomaly, i.e., loss of $CO_2$ to the atmosphere, due to enhanced respiration and
inhibited growth during that period (**Figure 3**c). The inversion sees this event in the
observations as well, suggesting even more $CO_2$ release than Biome-BGC across the South
Island. Unfortunately, a prolonged data gap in the LAU time series during that period caused





by a lack of field standards (**Figure 1**), leads to weaker constraints from the atmospheric $CO_2$
data and therefore larger uncertainty in the flux estimates in the South Island.
A signal of excess $CO_2$ release in February-March 2013 is seen by the inversion across the
North Island, too (**Figure 10**). The limited coverage of some areas in the North Island,
especially in the north and east (**Figure 9**), leads to high annual mean flux uncertainty for
individual regions and prevents a robust analysis as to which regions responded the most
strongly to the drought. Eddy covariance data from a dairy pasture site in the northwestern
North Island during a ~100 day drought in 2008 found a temporary loss of $CO_2$ to the
atmosphere, but the ecosystem recovered to become a net sink of $CO_2$ for the year (Mudge et
al., 2011).
**7.3   Annual fluxes**
The geographic air-land flux distribution averaged over 2011-2013 is shown in **Figure 11**,
including flux gradients on the sub-regional level that were prescribed in the inversion. A
comparison to the *a priori* distribution in **Figure 3**b shows larger areas acting as a net carbon
source. These include the central and north-eastern parts of the South Island, which roughly
correspond to the Canterbury region and mostly contain pasture land, in particular sheep and
beef pasture (**Figure 3**a). The inversion does not, however, resolve ecosystem processes, but
merely estimates net air-land fluxes, so it is not possible to make a link between pasture and a
net $CO_2$ source. A counterexample is the south-east of the South Island, which also contains
large areas of pasture, but is estimated to be a net carbon sink. In general, the inversion
assigns much more of the total land sink to forested areas than the Biome-BGC prior. This is
particularly apparent along the western South Island, but also in the eastern half of the North
Island. The strong flux gradients seen in region #3 are likely to be the result of the very
heterogeneous composition of LCLU types there, combined with BHD and LAU having low
sensitivity in the region (**Figure 9**), rather than a real signal. The inclusion of an additional
station with high sensitivity to the northern North Island, such as RBM, would be needed to
improve flux estimates there.
In the inset of **Figure 11**, we compare annual mean results from the inversion with bottom-up
estimates from the National Inventory Report (MfE, 2015), or NIR. The inversion suggests a
much larger net $CO_2$ sink across the country compared to the NIR. Particularly in the forest of
the south-western South Island, the inversion suggests both stronger photosynthetic and
respiratory activity than the prior model, with the overall balance towards a larger $CO_2$ sink



over the course of a year. For example, Fiordland appears to take up between 22 and 68 Tg
$CO_2$ each year in 2011-2013 (**Table 1**), which corresponds to per-area uptake rates of 614 and
1899 g $CO_2$ m$^{-2}$ yr$^{-1}$, respectively. By comparison, the Biome-BGC estimates range from 0 to
3 Tg $CO_2$.
The NIR estimates do not come with an overall uncertainty, but based on their reporting of
typical uncertainty for individual ecosystems, and personal communication, an approximate
figure of 50% was identified. This implies statistical significance for the difference in annual
sink estimates, except in 2013, when both estimates agree within their uncertainty range.
Without ocean prior or by assuming a baseline bias of 0.1 ppm, the differences are reduced by
up to a half (section 7.4), but do not disappear. How can these differences be explained?
There are a number of possible scenarios, which we explore in the following.
The accounting of fossil emissions differs between the NIR and the inversion. The EDGAR
emissions of 47.8 Tg $CO_2$ yr$^{-1}$ prescribed in the inversion contain elements of land-use change
and agriculture, which will therefore not be part of the posterior flux estimates. The NIR gives
total emissions of 34.6 Tg $CO_2$ yr$^{-1}$ for 2013. The difference of about 13 Tg $CO_2$ yr$^{-1}$ would
appear in the inversion as an additional sink of equal size.
The inversion and NIR estimates are not directly comparable, due to differences in the top-
down versus bottom-up viewpoints. While the inversion sees the overall net $CO_2$ exchange
between the atmosphere and the land, the NIR estimate represents the so-called LULUCF
sector, i.e., it includes contributions from Land-Use, Land-Use-Change and Forestry. In the
LULUCF model, it is assumed that $CO_2$ emissions from harvested wood products occur at the
location of the tree, a process particularly important to forest plantations located in the central
North Island and in the north of the South Island (**Figure 3**a). However, about 70% of the
biomass from forest harvesting is exported before major processing, i.e., in the form of logs,
sawn timber, or manufactured wood products (Pike, 2014). Most of the $CO_2$ release
associated with harvesting will subsequently occur far away from New Zealand, e.g., in China
with a 34% share of New Zealand's forestry exports in 2012. In a regional inversion these
emissions cannot be seen (unless being transported back into the inversion domain much later,
as part of the background concentration), leaving a larger net sink. Emissions from harvested
wood products are reported in the NIR at 10.3 Tg $CO_2$ yr$^{-1}$ in 2013, translating into about 7 Tg
$CO_2$ yr$^{-1}$ that cannot be seen by the inversion when assuming a 70% export rate. No emissions
are reported for earlier years, because the harvested wood products category was introduced





for the first time in the 2015 report. The 2013 estimate is likely to be an upper bound for the
years 2011 and 2012, because the volume of harvested wood products has increased steadily
since 2009 (MfE, 2015). Other possible discrepancies between the NIR methodology and the
net $CO_2$ fluxes for forests include the variance in the timing of root carbon emission following
tree mortality (Kirschbaum et al., 2013). Large sinks observed but not accounted for in NIR
can result from applying steady state assumptions to natural or pre-1990 forests when they are
accumulating carbon in biomass during recovery from past disturbance, with potential rates of
biomass accumulation by native species reaching 700-900 g $CO_2$ m$^{-2}$ yr$^{-1}$ (Trotter et al., 2005).
For pastoral agriculture, a more complex set of differences applies to the intercomparison of
inversion results, *a priori* process-based model results and the NIR methodology. Similar to
forestry, agricultural exports (e.g. milk, meat and wool) equated to 340 g $CO_2$ m$^{-2}$ yr$^{-1}$ for a
dairy pasture (Mudge et al., 2011), and the 165 Gg of nitrogen estimated as exported in
produce (Parfitt et al., 2006) will equate to an apparent net $CO_2$ uptake of 5.8 Tg $CO_2$ yr$^{-1}$
across New Zealand. The second-most important gap between methodologies results from the
NIR calculation that 6.3-6.5% of the energy content of pasture consumed by ruminants is
converted to $CH_4$ emissions. These $CH_4$ emissions represent a carbon flux to the atmosphere
not observable as $CO_2$, and therefore require separate quantification. They have been
calculated as 79 g $CO_2$ m$^{-2}$ yr$^{-1}$ in a dairy pasture (Mudge et al., 2011) and the carbon content
of the NIR's 1137 Gg of $CH_4$ emissions equates to an unobserved 3.1 Tg $CO_2$ yr$^{-1}$. Several
additional terms, including leaching of dissolved carbon forms, and imports of feed and
fertiliser can also provide important corrections between net ecosystem productivity (NEP)
seen by inversions and eddy-covariance and net ecosystem carbon balance (NECB) (Mudge et
al., 2011).
The NIR methodology also does not account for above or below-ground grassland biomass,
nor does it account for soil carbon changes. The process-based model Biome-BGC potentially
accounts for both these flux terms, but not in relation to intensive management. Therefore,
both biomass and soil carbon must be considered to explain additional $CO_2$ uptake or loss by
pastures that might be seen by the inversion, but not by NIR or Biome-BGC.
Biomass carbon is relatively small in New Zealand pastures (Tate et al., 1997), but can be a
significant component of seasonal net exchange (Mudge et al., 2011; Rutledge et al., 2015;
Hunt et al., 2016) as described in section 7.1. Repeated measurements of soil profiles suggest
that soil carbon changes can also be significant but uncertain due to limited sites available for



resampling. A recent analysis of all sites available nationally suggests that sites on flat pasture
are losing soil carbon at rates of ~170 g $CO_2$ m$^{-2}$ yr$^{-1}$, while sites in hill country are gaining
~770 g $CO_2$ m$^{-2}$ yr$^{-1}$ (Schipper et al., 2014). In addition to large areas of grazed pastures on
both islands, significant areas of tussock grasslands on the South Island could be gaining
biomass and soil carbon as they recover from historic overgrazing (Tate et al., 1997). The
extensive area of grasslands on both islands could result in large net $CO_2$ exchange fluxes
usefully observed by inversion studies. New Zealand's first process-based studies of net
national ecosystem carbon balance suggested large uncertainties in grasslands (Tate et al.,
2000) and later suggested grasslands were approximately carbon neutral in 2001 (Trotter et
al., 2004). Eddy covariance studies remain limited in coverage across New Zealand, but tend
to suggest potential for large negative NEP and near neutral NECB. Rutledge et al. (2015)
updated and extended the Mudge et al. (2011) results to 4 years, yielding average NEP of
600±180 g $CO_2$ m$^{-2}$ yr$^{-1}$ and NECB of 220±200 g $CO_2$ m$^{-2}$ yr$^{-1}$. Hunt et al. (2016) also report
eddy-covariance carbon budgets for an irrigated intensively-grazed dairy pasture and an
unirrigated winter-grazed pasture in Canterbury on the South Island's east coast. Over one
year, the unirrigated pasture was carbon neutral (±80 g $CO_2$ m$^{-2}$ yr$^{-1}$), while the intensively-
managed and irrigated pasture displayed NEP of 1500±140 g $CO_2$ m$^{-2}$ yr$^{-1}$ and NECB of 380
(±150) g $CO_2$ m$^{-2}$ yr$^{-1}$.
The forest and grassland studies described above suggest that large, negative flux anomalies
estimated by the inversion may be plausible when extrapolated across the large areas of these
LCLU categories. It is important to remark that the inversion will see estimates similar to
NEP, but that eddy covariance studies have demonstrated that NEP can be corrected to NECB
using NIR-compatible data without the introduction of larger errors.
Additional real land carbon balance terms may also contribute to large, negative flux
anomalies that differ from NIR and process-based models such as Biome-BGC. These terms
include areas of organic soil accumulation in wet forests and bogs, typical of west-coast
environments where the largest negative flux anomalies are observed. Campbell et al. (2014)
used eddy covariance to find NEE of 800–900 g $CO_2$ m$^{-2}$ yr$^{-1}$ with a strong seasonal cycle.
Erosion and deposition can also create a net carbon sink that may be unusually significant in
active margins such as New Zealand (Tate et al., 2000; Baisden and Manning, 2011). Small
catchment and site studies have estimated rates of net pasture soil carbon sequestration due to
erosion and burial, accounting for upland soil carbon recovery, of 220 g $CO_2$ m$^{-2}$ yr$^{-1}$ (Page et





al., 2004) and 370 g $CO_2$ m$^{-2}$ yr$^{-1}$ (Parfitt et al., 2013). Scott et al. (2006) have estimated the
national delivery of eroded carbon to the coast as 11±4 Tg $CO_2$ yr$^{-1}$ and suggested that much
of this carbon is likely to be buried and replaced in uplands. Dymond (2010) attempts to more
fully and dynamically account for erosion, burial and replacement, suggesting a range of 4-20
Tg $CO_2$ yr$^{-1}$. Both studies suggest the largest erosion-induced $CO_2$ sinks occur in the Southern
Alps in the west of the South Island, where the Lauder station allows observation of a strong
sink, as well as in the North Island's east coast. These estimates may partly be included in the
hill country soil carbon accumulation estimated by Schipper et al. (2014). Smith et al. (2015)
suggest that fiords may also create a strong carbon sink, with about 18 Mt of organic carbon
being buried in fjord sediments globally each year, yielding a rate of 198 g $CO_2$ m$^{-2}$ yr$^{-1}$.
Thus, a number of plausible suggestions have been documented in the literature to help to
explain why the $CO_2$ sink seen by the inversion is stronger than estimated in the NIR.
**7.4   Uncertainty and bias assessment**
In addition to regional uncertainty, the posterior covariance matrix from the Bayesian
optimization also contains spatiotemporal error correlations between regions and every week
within each region. These correlations are fully taken into account when reporting
uncertainties for aggregated regions. Strong negative correlations between two regions would
indicate that the inversion is unable to distinguish their individual flux components with the
available data, but only their sum. Similarly, positive correlations are indicative of the
difference of flux components being constrained better than each individually. An analysis of
the error correlations reveals that both negative and positive correlations are present, however,
only 0.13% of all pairwise correlations have an absolute value greater than 0.1. Very few
values are smaller than -0.4 or greater than 0.2, with the negative extreme around -0.7 and the
positive extreme around 0.3. Hence, with the available data, the inversion appears able to
resolve weekly fluxes on the regional level chosen.
An analysis of the mismatch of modelled $CO_2$ (i.e., the $CO_2$ time series obtained by
propagating the posterior flux through the transport model) and observed $CO_2$ reveals
differences between the BHD and LAU stations (**Figure 12**). At BHD, the mismatch
distributions are very similar for the 13:00-14:00 and 15:00-16:00 time series, have a bias of
11-13% of the prior data uncertainty and show no discernible temporal pattern (smoothed,
thick lines in the figure). At LAU, the mismatch distribution is similar for the 13:00-14:00
time series, with an even smaller bias of 7%, but for the 15:00-16:00 time series there is a





much larger bias of 52% of the data uncertainty. This means the inversion has difficulties
reproducing the low $CO_2$ concentrations in the LAU 15:00-16:00 observational record. The
temporal evolution indicates an alternating pattern of small mismatch during the (austral)
winter and larger mismatch during summer.
One possible explanation is that the representation of the planetary boundary layer (PBL) in
the model is too shallow in the late afternoon during summer. At a site like LAU, strong solar
radiation during a clear summer day might lead to a sudden deepening of the PBL in the
afternoon between the two release periods, which might not be fully captured by the NZLAM
meteorology. If the real PBL is deeper than in the model, any signal from surface fluxes
would be mixed in a larger volume of air and measured $CO_2$ concentrations would be lower
than what is assumed by the model. The inversion would have difficulties matching these
lower concentrations and end up with a positive bias.
We compared the model boundary layer depth at 15:00-16:00 to radiosonde measurements
made at LAU (**Figure 13**). The Heffter method (Heffter, 1980) was used to compute PBL
height from the radiosonde data. The comparison suggests that the boundary layer is indeed
too shallow in the model during summer. However, this comparison has caveats, because the
radiosonde dataset is preliminary and only few measurements were taken during the right
time of day (15:00-16:00 LT). Furthermore, an equivalent analysis with the 13:00-14:00 LT
data suggests a similar discrepancy, so the question remains why these data can be explained
by the inversion, yet not the 15:00-16:00 LT data.
Results from the sensitivity scenarios (i)-(iii) are incorporated in the figures as an additional
uncertainty band on top of the Bayesian posterior uncertainties from the default run (**Figure**
**10**). That band represents the maximum (minimum) value of the flux plus (minus) its
uncertainty at every point in time and across all runs, i.e., including the default and sensitivity
runs.
While the uncertainty range associated with the suite of sensitivity scenarios is symmetrical
around the reference case for most regions, the sensitivity range is characterised by more
positive flux estimates than the reference case in the western South Island (**Figure 10**), i.e., a
slightly smaller annual carbon sink. This can be attributed to sensitivity case (iii), in which the
inversion is allowed to adjust air-sea fluxes to any value without penalty. Some of the
terrestrial $CO_2$ uptake is relocated to upwind ocean regions, as this yields a lower Bayesian
cost, because fluxes from the western South Island are shifted towards the Biome-BGC prior



estimates. However, in order to offset a relatively small flux change on land, the change in
ocean flux has to be large due to the distance to the stations and the dilution of $CO_2$
concentrations on the way. This leads to an ocean sink of 6 Pg $CO_2$ in 2012 in our regional
domain for this sensitivity test, which is more than ten times larger than estimates for the
whole Southern Ocean from global inversions, ocean carbon data, and ocean biogeochemistry
models (Gruber et al., 2009). Despite this unrealistic result for the oceans in the sensitivity
test, the conclusions about the seasonal pattern in $CO_2$ uptake and release and its spatial
distribution in the New Zealand land regions remain robust.
The inversion assumes an unbiased baseline $CO_2$ record. Any positive (negative) bias would
be interpreted by the inversion as an additional sink (source) of $CO_2$. From the sensitivity runs
we find that a constant bias in the baseline of 0.1 ppm would cause the total $CO_2$ flux of New
Zealand for each year in 2011-2013 to be off by approximately 20 Tg $CO_2$ $yr^{-1}$. This
corresponds to about 50% of the flux uncertainty from the default run (**Table 1**), thus
underscoring the importance of an accurate baseline in a regional inversion. Similar to the
sensitivity case without ocean prior, the biased baseline has only a minor influence on
seasonal flux patterns over land. An inversion such as ours can always benefit from advances
in air-sea flux datasets, such as $pCO_2$ measurements from a regional cruise network, as well
as well-characterized background air concentrations. One way to aid the baseline
representation in future top-down studies of the New Zealand region could be to add a
western component in addition to the southern and northern components, which would
improve the characterization of air that carries an Australian signal on top of the Southern
Ocean background. This could be accomplished, e.g., by establishing a $CO_2$ measurement
station situated along the west coast of South Island and choosing a western background
sector.
**8   Conclusions**
We present the first regional inversion estimates of air-land and air-sea $CO_2$ fluxes for the
New Zealand region, which were estimated from two *in situ* observing stations in New
Zealand, ship based measurements, and Lagrangain model simulations using the NAME
dispersion model driven by NZLAM meteorology. The results imply a strong seasonal cycle,
especially for fluxes in the western South Island. Regions covered predominantly by
indigenous forest appear to have more pronounced photosynthetic and respiratory activity





than suggested by the land model. This is most apparent in Fiordland, which is a key
contributor to the seasonal cycle, as well as the annual mean sink, in the South Island. The
timing, magnitude and regional distribution of seasonal flux patterns are well constrained and
robust across sensitivity cases, while uncertainties in annual totals are more significant.
Enhanced $CO_2$ release from the terrestrial biosphere in New Zealand is apparent in response
to the 2012/2013 drought period. This response appears most prominent in the North Island
and western parts of the South Island, consistent with reports about these regions being most
severely affected.
The annual total $CO_2$ sink in New Zealand is estimated to have decreased over the 3-year
period, at 132 ±36, 97 ±36 and 64 ±40 Tg $CO_2$ yr$^{-1}$ in 2011, 2012 and 2013, respectively. The
New Zealand national inventory reports a much smaller sink of 28, 27 and 27 Tg $CO_2$ yr$^{-1}$ for
the same years (with uncertainty around 50%). About 7 Tg $CO_2$ yr$^{-1}$ of the discrepancy can be
attributed to emissions associated with forest harvesting, which are included in the inventory
but missed by the inversion due to forestry exports. Another 13 Tg $CO_2$ yr$^{-1}$ arise from
different accounting of fossil emissions between the inventory and the inversion. Additional
factors relating to the difference between NEP and NECB in pastures can account for another
9 Tg $CO_2$ yr$^{-1}$, Other terms such as erosion, burial and soil carbon recovery may account for
another 4-20 Tg $CO_2$ yr$^{-1}$. These differences largely reconcile both results for 2013, but not
2011-2012. Carbon sequestration by grassland and soil carbon could also play an important
role in causing differences between the two methods, as these processes are not included or
fully resolved in inventory reporting but would be seen by the inversion. Collectively, these
factors are likely to reconcile both results only partially, with some differences remaining.
Detailed sensitivity studies suggest that the most important causes of uncertainty in the
inverse estimates are uncertainties in the estimate of baseline air entering the domain and air-
sea fluxes from the ocean surrounding New Zealand. These uncertainties could be reduced
through more dense $pCO_2$ measurements in the oceans around New Zealand, and extending
the ship based atmospheric $CO_2$ measurements presently used to estimate the baseline air
farther to the south and west. Another possibility is to establish additional surface stations in
strategic locations, i.e., with footprints in areas where Lauder and Baring Head have low
sensitivity, such as Rainbow Mountain in the North Island, or along the west coast of the
South Island.



The inversion methodology developed here is a powerful tool to validate net regional $CO_2$
sinks in the New Zealand national inventory report. It offers an independent, top-down view
on the national carbon budget.
**Appendix A  -  Lauder site description**
*A.1   The Lauder station*
The Lauder atmospheric research station (45.038S, 169.684E, 370m AMSL) is located in the
broad Manuherikia river valley on the South Island of New Zealand. A semi-arid continental
climate predominates with an annual rainfall of 450mm and mean annual temperature of
9.7C. The prevailing wind is from the westerly quarter (a mean daily wind run of
approximately 300 km). Periodic southerly frontal systems bring air masses from the
Southern Ocean and Tasman Sea. The research station is located 35 km north of the township
of Alexandra (population: 5000). The station is surrounded by pastoral land dominated by
sheep and cattle farming practices along with seasonal cropping. Farming practices are non-
intensive and stock numbers are relatively low. The valley is sparsely populated. The land
westward (upwind) of the valley consists of numerous valley systems and mountainous
terrain. The vast majority of this land is undeveloped and is part of New Zealand's national
park system. There is no major industry present in the region.
Due to the relatively clear unclouded skies, low light pollution and low levels of local and
regional anthropogenic emissions, 'clean air' ground-based remote sensing, balloon sonde and
*in situ* measurements are routinely conducted at the station as part of NDACC (formely
known as NDSC) (Kurylo, 1991), GAW (WMO-GAW, 2007), TCCON (Wunch et al., 2011)
and GRUAN (Seidel et al., 2009) activities.
*A.2   Lauder in situ trace gas measurements*
Long term routine *in situ* measurements began at Lauder in 2003 with the installation of a
TEI-49C Ozone monitor (Zellweger et al., 2010). Previous to this only sporadic short term
campaigns focusing on tropospheric nitrogen dioxide had been undertaken (Johnston and
McKenzie, 1984). Continuous *in situ* measurements of carbon dioxide ($CO_2$), methane ($CH_4$),
nitrous oxide ($N_2O$) and carbon monoxide (CO) began in March 2007 when a prototype FTIR
trace gas analyser was installed (Griffith et al., 2012; Sepúlveda et al., 2014). In June 2008 a
well-calibrated continuous $CO_2$ NDIR (differential, non-dispersive, infrared) analyser (LI-





7000, manufactured by LI-COR, Inc, USA, www.licor.com) was installed at Lauder. This was
followed by regular fortnightly flask samples analysed for $CO_2$, $CH_4$, $N_2O$, CO and $\delta^{13}C$-$CO_2$
concentrations, starting in May 2009. An added advantage of employing the NDIR analysers
at both sites (Lauder and Baring Head) is that they share common data processing code and
calibration routines.
*A.3   Air inlet system*
The air inlet system consists of a permanent 10 m high NIWA meteorological mast erected at
a distance of 33 m, to the north, from the nearest building (which also houses the *in situ*
instrumentation). The meteorological mast is constructed with metal irrigation piping. Two
sets of 60 m long (ID 8.8 mm) baked copper tubing were used to collect air from the mast
(inlets located at 6 m and 10 m) and deliver it to two distribution manifolds (one for each
sample line). A custom made inverted funnel with coarse mesh (0.7 mm) is used to provide
inlet rain and dust protection. In June 2012 the copper sampling lines were replaced with
stainless steel (SS) tubing (ID 8.8 mm). Sampling manifolds are inserted into the sampling
lines next to the instruments. A 100 mm long segment of PFA 9.5 mm tubing is inserted
between the sampling lines and the manifolds to electrically isolate sampling systems and
instrumentation from the meteorological mast. The manifolds are constructed from 25 mm SS
diameter tubing 200 mm in length (volume = 0.086 l). Each port consists of a 6.3 mm SS tube
welded perpendicular to the main body. Each port extends 15 mm into the main body and
terminated with a 45 degree angle cut facing the direction of flow.
Sample air is drawn into the two 4-port manifolds with a roughing pump (KNF Neuberger,
N035 AN18) at 10-15 l min$^{-1}$ giving an effective residence time of approximately 35 seconds
and an associated pressure drop of 40 mbar. The roughing pump allows sample air to be
drawn at a higher flow rate and allows multiple instruments to be connected to the sample
lines without front end pressure coupling between co-sampling instruments.
The LI-7000 is connected to one of the 10 m sampling line manifold ports with 6.3 mm
Synflex© (Registered 1300) tubing. The LI-7000 inlet system extracts sample air from the
manifold at a rate of 2.6 l min$^{-1}$. An FTIR trace gas analyser draws sample air from a 10 m
sampling line manifold port through a 6.3 mm PFA tube (300 mm length) at a rate of 3.5 l
min$^{-1}$ and the flask sampling system is connected to the same 10 m sampling line manifold
port with 3.2 mm Ledalon© (1200 Series Nylon 12 Tubing) tubing. When flask samples are
taken a flow rate of up to 2 l min$^{-1}$ is used. Currently no measurements are taken on the 6 m
line.

### A.4   LI-7000

The LI-7000 is a commercially available dual cell NDIR analyser able to calculate $CO_2$ mole
fractions via measurements in the $CO_2$ 4.255 μm absorption band. The LI-7000 has been
proven to be a low maintenance robust $CO_2$ analyser able to meet GAW measurement criteria
when operated in the correct manner (WMO, 2001). A gas delivery and data acquisition
system designed by NIWA (Gomez, 1997) is used to automate and manage the delivery
sample and reference air along with calibration gas to the LI-7000. The LabView© data
acquisition program and hardware that is used to control gas delivery also performs data
management and display of real time instrument diagnostics. The Lauder gas handling and
data acquisition system is an earlier version of the current Baring Head continuous $CO_2$
monitoring system described in Brailsford et al. (2012). The main difference is that a LI-7000
is employed at Lauder whereas at Baring Head a Siemens Ultramat 3 gas analyser (M52012)
is used.
The LI-7000 draws air from the aforementioned air inlet system 10 m manifold via a
diaphragm pump (KNF Neuberger, KNF 86KNE, 2.6 Lmin$^{-1}$). A set of four Field standards,
with a calibration lineage to the mole fraction scale maintained by the CCL and a
target/archive tank are connected to a valve manifold consisting of five three-way (Parker
B16DK1175) valves in a daisy chain configuration, along with the dried air allowing selection
of either Field standards, target tank or sample air for the analyser. Gas regulators (Scott
Marin Inc, 1-SS30-590-DAT) and 1.6 mm SS tubing are used to connect tanks to the gas
delivery system. Compression fittings (Swagelok©) are employed for all connections.  On the
outlet of the sample pump an overpressure is maintained on the inlet to a Nafion drier (Perma
Pure inc, MD-110-144S-4) with the excess flow vented, this removes the bulk of the water
content from the sample flow. The air sample then passes through a magnesium perchlorate
trap to ensure all gas to be measured has the same low water content before being introduced
to the analyser by a 100 sccm mass flow controller (McMillian, 80SD-5). One of the Field
standards is also used in the reference cell as a reference gas, and is controlled using a similar
mass flow controller (McMillian, 80SD-3) at 10 sccm. The exhaust sample and reference gas
are then dried again on molecular sieve trap before acting as the counter flow on the Nafion
drier, in this way dew points of -65 C are consistently met.





The data acquisition system selects the calibration gas to measure and monitors each Field
standard for stability to optimise the gas consumption. When a Field standard has a standard
deviation of less than 0.015 ppm over a minute it is defined as stable and the next gas is
measured. Sample air is continuously measured with 5-minute averages collated and reported.
Interspersed at regular half hourly intervals, individual Field standard tanks are measured.
Every 4-6 hours the suite of four Field standards is measured. A target/archive tank is
measured every 23 hours. Each week the Field standards and target tank are measured as a
separate aliquot multiple times. This sampling sequence is akin to the calibration protocol
employed by Brailsford et al. (2012) and Stephens et al. (2011). Data processing is performed
by Lauder LI-7000 specific scripts adapted from those used by (Stephens et al., 2011) and
written in the free statistical analysis software R.
Allan variance measurements (Allan, 1966) show the precision of the coupled LI-7000 -
NIWA gas delivery system as 0.004 ppm (1 sigma in five minutes). Calibration of the LI-
7000 is obtained by fitting a 3rd order polynomial to the measurements of the four Field
standards to characterise the concentration dependent nonlinear response of the instrument
every 4-6 hours. This calibration curve is then used to calibrate sample air measurements,
putting the measurements on the WMO X2007 scale. $CO_2$ concentrations of the Field
standards are constructed to evenly span the typical air sample concentration range
encountered, including elevated nocturnal levels (typical span of 380-450 ppm). Thirty
minute zero offsets are calculated using the interspersed individual Field standard
measurements. Instrument dependent artefacts (e.g instrument temperature and flushing
times) are accounted for in the processing code by calculating a linear fit of known Field
standard concentrations and the parameter in question.
The Field standards and the target tank are filled at BHD and characterised at the NIWA
GASLAB, Greta point, Wellington, New Zealand. The Field standard $CO_2$ concentrations are
calibrated to the WMO X2007 scale, along with $\delta^{13}C$-$CO_2$ (PDB-AIR3.3 scale) (Brailsford et
al., 2012). Field standards require changing every 12-18 months. The target tank requires
changing every 6-12 months as in parallel it also functions as a target tank for the FTIR trace
gas analyser.
***A.5   Meteorological sensors***
Meteorological sensors were installed onto the sampling mast. Wind speed is measured at
three heights (2.8 m, 5.8 m and 10.1 m) using Vector instruments A100LK anemometers. A





Vector instruments W200P wind vane mounted at a height of 10.1 m is used to record wind
direction. Relative humidity and temperature are measured using Vaisala Humitter 50U/50Y
sensors, placed at heights of 2.6 m and 9.9 m. In addition, a Vaisala PTB100 analog
barometer was installed adjacent to the *in situ* instruments (inside the building). All these
sensors are connected to a Campbell CR10X data logger and SDM-INT8 logger module. Ten-
minute averages of all sensor output are recorded independently of *in situ* gas measurement
instrumentation output.
**Appendix B  -  Baseline Analysis**
A $CO_2$ baseline is constructed as a weighted average of a southern and northern baseline,
which takes into account whether the modelled trajectories originated to the north or south of
the inversion domain for a given data point.
*B.1   Southern Baseline*
The southern baseline represents a continuous record of steady background $CO_2$ mole
fractions during southerly wind conditions at BHD. A multi-step filter is applied to the BHD
record to obtain a $CO_2$ baseline representative of a large region over the Southern Ocean, as
described by Brailsford et al. (2012) and Stephens et al. (2013). In short, the filter selects
measurements during extended periods of southerly winds at the site, during which a
maximum standard deviation of 0.1 ppm is achieved. Additional meteorological conditions
must be fulfilled to preclude the influence of local sources and to ensure the air has not passed
over the South Island before arriving at BHD. The result of this filtering process is similar to
selecting observations from the southern cluster in **Figure 7**. The southern baseline based on
the filtering is used in this study.
After filtering the data for baseline conditions, a continuous baseline is constructed using the
seasonal time series decomposition by Loess (STL) algorithm (Cleveland et al., 1990), which
allows estimation of a long-term trend and interannually varying seasonal patterns. The STL
algorithm uses two time windows for the seasonal cycle and the trend, which are set by the
user and define the respective time periods over which variations in the data are considered.
The monthly averaged data fulfilling the baseline conditions are used as input, and the
algorithm is run first with a seasonal cycle window of 5 years and a trend window of 121
months to single out the decadal trend. This trend is then removed before a second run with a





trend window of 25 months to capture the interannual and seasonal patterns. Finally the
decadal, interannual and seasonal time series are summed and the resulting baseline
subsampled at the 13:00-14:00 and 15:00-16:00 LT windows.
### *B.2   Northern Baseline*
The northern baseline is based on *in situ* $CO_2$ observations using a NDIR analyser on board
the TF5, a ship of opportunity that cruised the triangle Japan/Australia/New Zealand about
once a month during the period 2011-2013 (Chierici et al., 2006). The cluster analysis showed
that during northerly events the air is usually coming from the northern Tasman Sea or the
subtropical waters to the north and only occasionally from the South Pacific eastward of New
Zealand. The layout of the TF5 cruises with legs crossing the Tasman Sea as well as
subtropical legs therefore offers the possibility to characterise the $CO_2$ concentrations in these
regions with monthly resolution. The TF5 dataset provides $CO_2$ concentrations averaged over
10 minute intervals along with the standard deviation of the high frequency measurements
within these intervals.
We defined a regional mask (**Figure 5**) to keep observations from the open ocean and avoid
observations taken close to the land, especially near the Australian east coast as it is located
upwind during average south-westerly conditions and hosts large urban centres with
significant $CO_2$ emissions. The mask spans the latitudes 39 S to 24 S. The mask was then
further partitioned into bands spanning one degree of latitude and the data within each band
averaged for each month in 2011-2013. The uncertainty of the resulting monthly record is
taken as the quadrature sum of the standard deviation of high-frequency data points measured
during the ship's transit of the respective latitude band and the standard deviation of the 10
minute data about the monthly mean.
The monthly record within each latitude band was analysed using the same STL routine as for
the southern baseline. An overall uncertainty estimate was formed by root mean square
combination of the monthly uncertainty and the time series of the remainder from the STL
analysis. The remainder time series is the difference between the monthly record and the sum
of the seasonal and trend components from the STL analysis. Finally, the STL baseline for the
two latitude bands for 27-26 S were averaged to produce a baseline representative of the
northern edge of the inversion domain at 26 S.
### *B.3   Weighted Baseline*



A day-to-day baseline was constructed as a weighted superposition of the southern and
northern baselines, with weights depending on the proportional latitude of air origin for the
twice-daily measurements at BHD and LAU. The daily NAME station footprints for the
13:00-14:00 and 15:00-16:00 LT windows were integrated along the southern and northern
edges of the domain to determine the relative fraction of back-trajectories leaving the domain
to the south and north. These fractions are then used to weigh the two baselines and create a
baseline associated with each of the twice-daily data points. Uncertainties are weighed in the
same way.
For most days the 13:00-14:00 and 15:00-16:00 LT footprints are similar with regard to the
origin of the air, so the weighted baselines for both time windows are almost identical. For
both stations on a typical day the region where the air originates is either clearly in the north
or the south, so that the weights for the southern and northern baselines are close to zero and
one, or vice versa. However, middle cases can occur when the wind conditions at a site
rapidly changed during the one-hour period over which measurements are collected, which
often results in two main branches of trajectories originating in the north and south
respectively. In this case both baselines are weighted proportionally to reflect the mixed
origin of air during the one hour averaging period for the measurements. Yet other days, or
periods of days, are characterised by slow wind speeds, sometimes slow enough that most
back-trajectories end before reaching either the northern or southern edge of the domain. In
this case, the midpoint of the footprint is determined and its latitude used to proportionally
weigh the baselines. The same procedure applies for days with trajectories leaving the domain
predominantly to the west or the east.
**Acknowledgements**
The author(s) wish to acknowledge the contribution of New Zealand eScience Infrastructure
(NeSI) to the results of this research. New Zealand's national compute and analytics services
and team are supported by the NeSI and funded jointly by NeSI's collaborator institutions and
through the Ministry of Business, Innovation and Employment. URL http://www.nesi.org.nz.
In addition, KNS, GB, DS, and SMF would like to acknowledge NIWA core funding through
the Greenhouse Gases, Emissions and Carbon Cycle Science Programme. We thank our
colleagues from New Zealand's Ministry for the Environment (MfE), especially the LUCAS





team, for very fruitful meetings. None of this work could have been accomplished without the
station operation teams at Baring Head and Lauder. We are grateful for access to radiosonde
data from Lauder, and would like to thank Ben Liley for his PBL calculations. We would also
like to thank Paul Wennberg and his TCCON team for the generous loan of their Li-7000 that
is currently at Lauder. The National Center for Atmospheric Research is sponsored by the
National Science Foundation.

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



1   **Figures and Tables**

3   ***Table 1***. *Annual mean $CO_2$ flux for selected aggregated regions, in Tg $CO_2$ $yr^{-1}$. A negative sign indicates uptake*
4   *by the land. In parantheses the a posteriori uncertainty ($1\sigma$) is shown (excluding the sensitivity cases).*

| Region | 2011 | 2012 | 2013 |
|---|---|---|---|
| NZ Total | -132 (36) | -97 (36) | -64 (40) |
| North Island | 18 (28) | -40 (28) | -1 (30) |
| North | 5 (25) | -10 (25) | 7 (25) |
| South | 13 (17) | -30 (17) | -8 (19) |
| South Island | -149 (22) | -56 (23) | -63 (28) |
| East | -37 (17) | 9 (18) | -10 (23) |
| West | -113 (17) | -65 (16) | -52 (17) |
| Fiordland | -68 (13) | -22 (12) | -31 (14) |

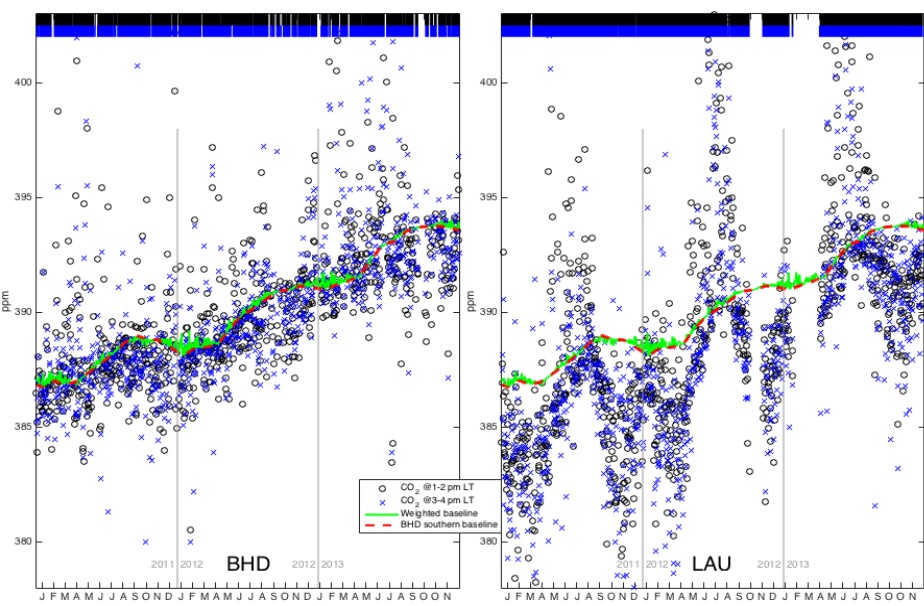

***Figure 1***. *$CO_2$ observations from BHD and LAU, twice-daily at 13:00-14:00 and 15:00-16:00 LT, through 2011-13. The BHD southern baseline is shown along with the weighted baseline used in the inversion. Gaps in the coloured bars at the top indicate days when no observations are available.*





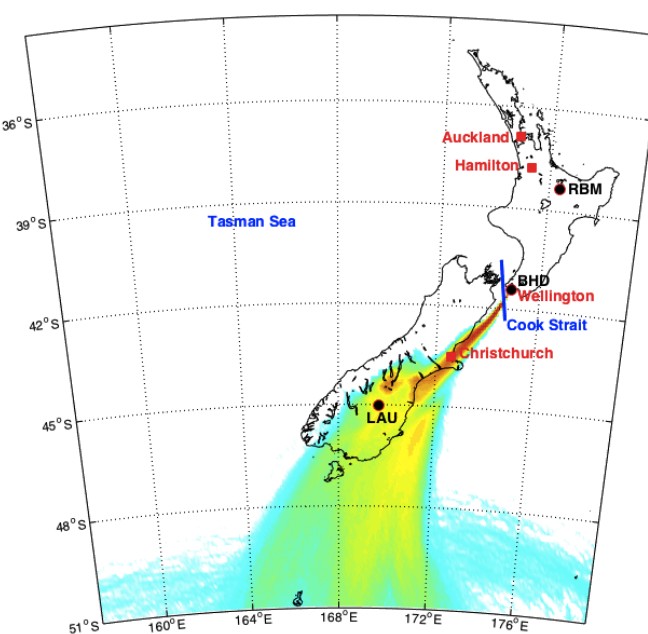

***Figure 2****. Air history map for 10,000 particles released at BHD during 15:00-16:00 LT on 19 May 2012 using the NAME III model. The particle back-trajectories show a southerly event locally at the station, though not a baseline event as the air crosses parts of South Island. Also shown are the locations of LAU and RBM stations.*





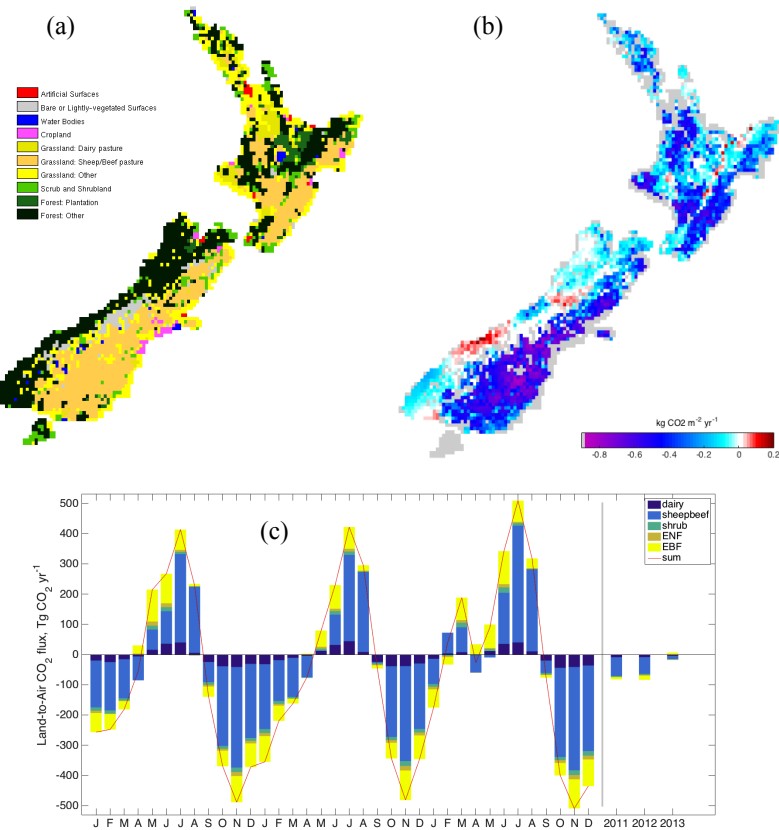

*Figure 3*. *(a) Land-Cover/Land-Use (LCLU) map with 10 categories, based on the Land Cover Database (LCDB3) and the Land-Use in Rural New Zealand (LURNZ) basemap. (b) A priori $CO_2$ flux distribution, averaged over 2011-2013, from modelled NEP using BiomeBGC with the LCLU categories. Both maps have been regridded to the NAME model grid. (c) Monthly and annual contributions from each biome to the overall a priori $CO_2$ flux.*



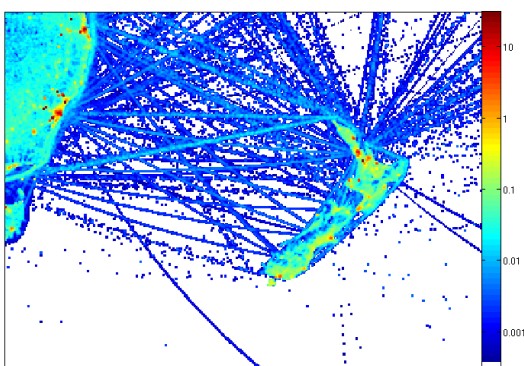

*Figure 4. Gridded fossil emissions of $CO_2$ across the model domain and averaged over 2011-2013, in kg $CO_2$ $m^{-2}$ $yr^{-1}$. Emissions are based on EDGAR v4.2, with 2011-2013 estimates extrapolated from the 2000-2010 trend in global total emissions.*

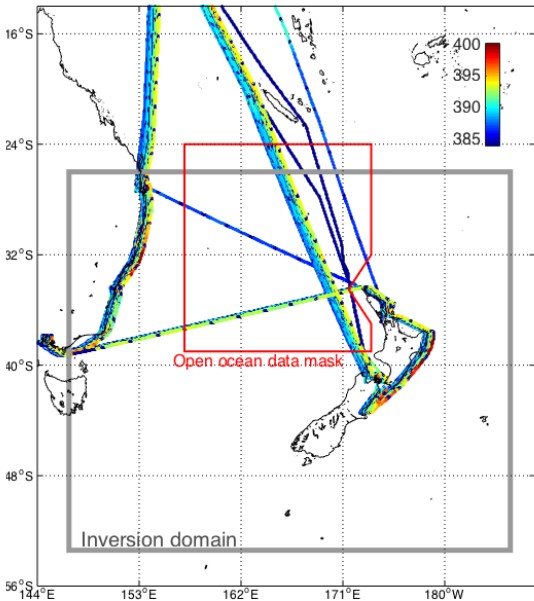

*Figure 5. Voyages of the Ship of Opportunity "Trans Future 5". Ship tracks have been slightly spread longitudinally to allow one to differentiate individual cruises (more recent cruises are to the right). Colors give the in situ $CO_2$ concentration. Measurements made inside the open ocean mask were used in the northern baseline analysis. The inversion domain is outlined in gray.*



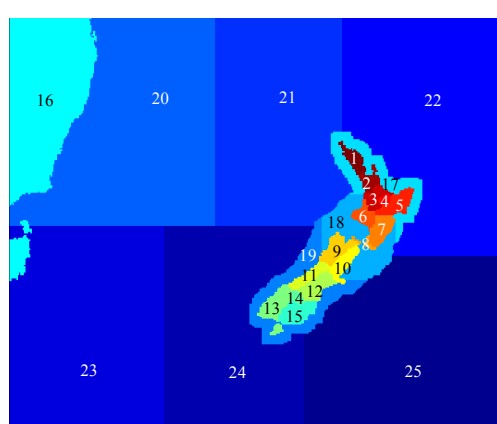

***Figure 6***. *Regional partitioning and indices in the inversion.*




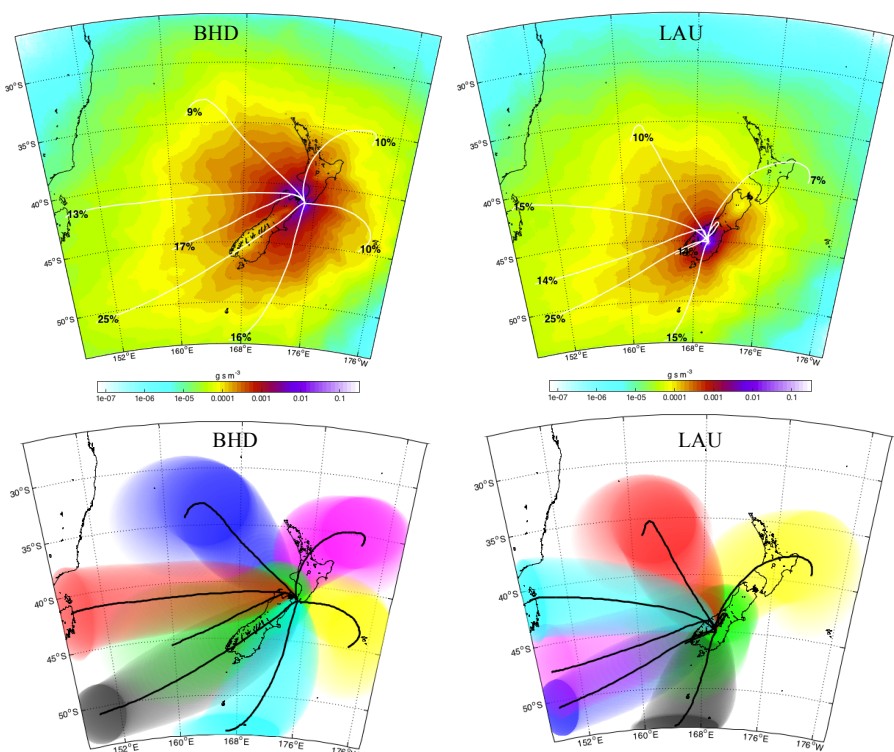

*Figure 7. Top panel: 2011-2013 mean footprints for BHD and LAU stations, based on twice-daily air history maps at 13:00-14:00 and 15:00-16:00 LT. Clusters of 4-day back-trajectories are overlain. Percentages give the sizes of clusters, i.e., the probability that a particle released on a random day has followed that pathway. Lower panel: Major atmospheric transport pathways for both stations from cluster analysis of the back-trajectories from twice-daily particle release. Shades represent the geographical spread of each pathway (one standard deviation from cluster centroid in latitude/ longitude).*





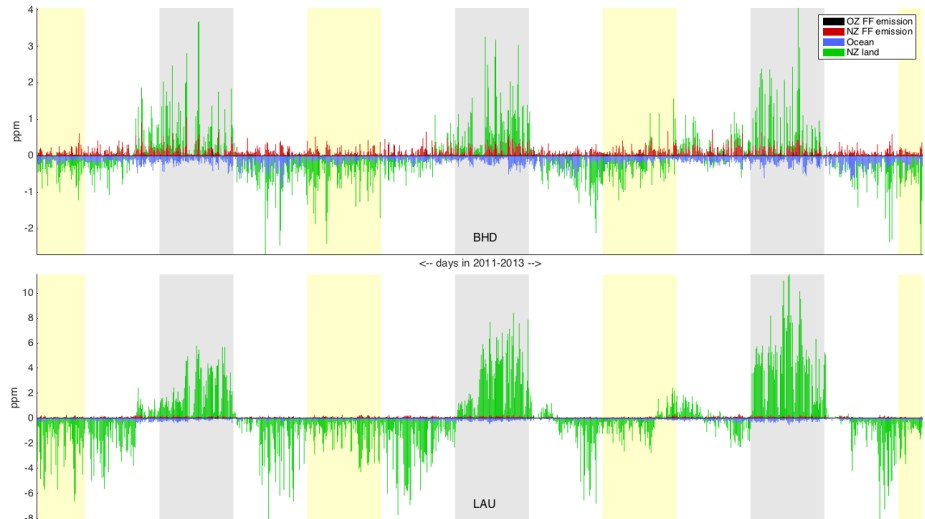

*Figure 8. Simulated contributions to the observed CO₂ anomaly, i.e., concentration minus baseline, at BHD (top) and LAU (bottom) for each day in 2011-2013 averaged over both 13:00-14:00 and 15:00-16:00 LT release periods. Contributions are calculated using NAME transport matrices with EDGAR v4.2 fossil fuel emissions, prior oceanic CO₂ flux from the Takahashi pCO₂ dataset and prior terrestrial CO₂ flux from the BiomeBGC model. Note that scales vary due to stronger anomalies and seasonal amplitude at LAU.*




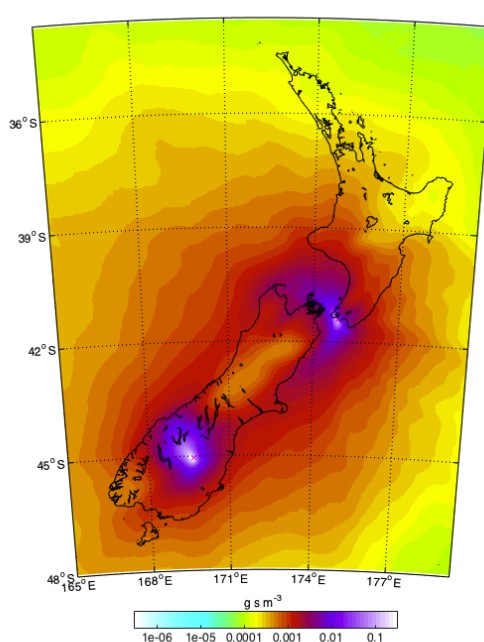

*Figure 9*. Combined 2011-2013 footprint for both 13:00-14:00 and 15:00-16:00 LT release periods for BHD and LAU around New Zealand.

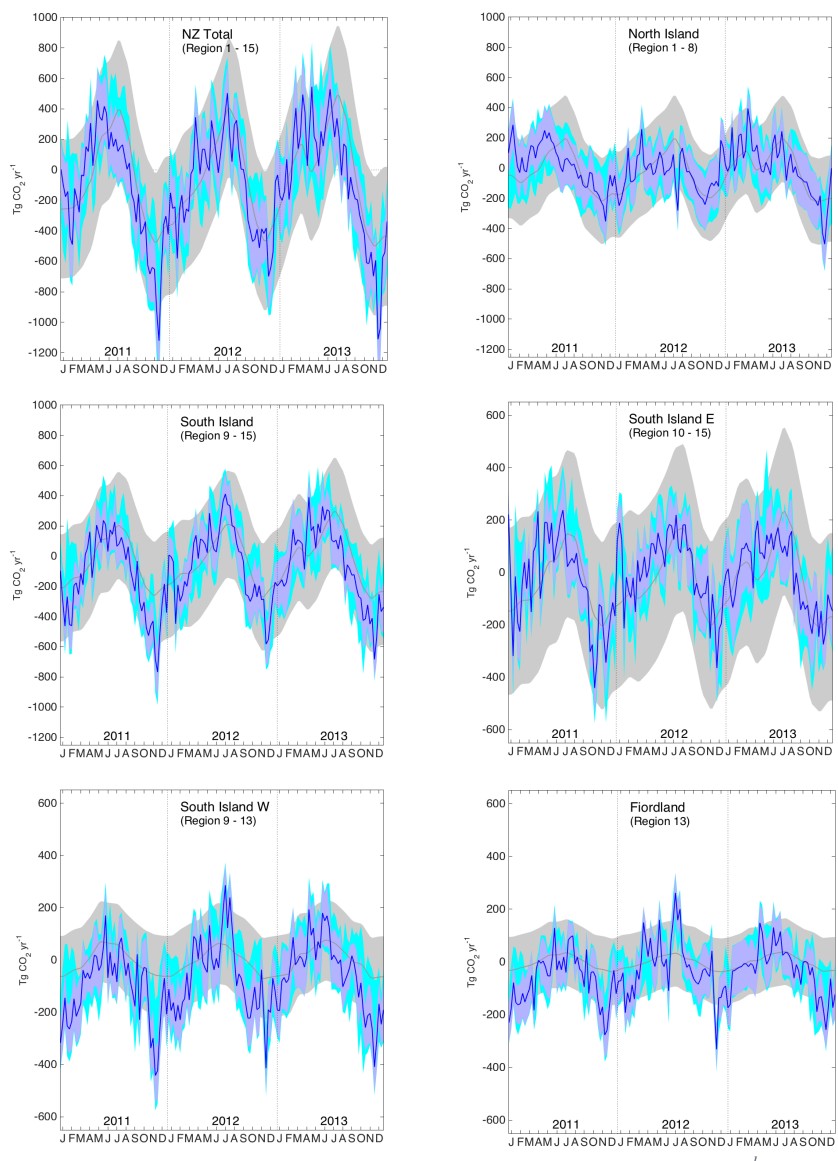

***Figure 10***. *Weekly CO$_2$ fluxes in 2011-2013 from selected regions, in Tg CO$_2$ yr$^{-1}$. Prior flux estimates are shown in gray and the inversion results are shown in blue. Shaded areas represent flux uncertainty (1σ). The cyan shade represents the extra uncertainty obtained from the sensitivity cases. Note there is a one-off change in scale of the flux axis for sub-island scale regions. A positive flux indicates a net release of CO$_2$ to the atmosphere, while a negative value indicates uptake by the land biosphere.*





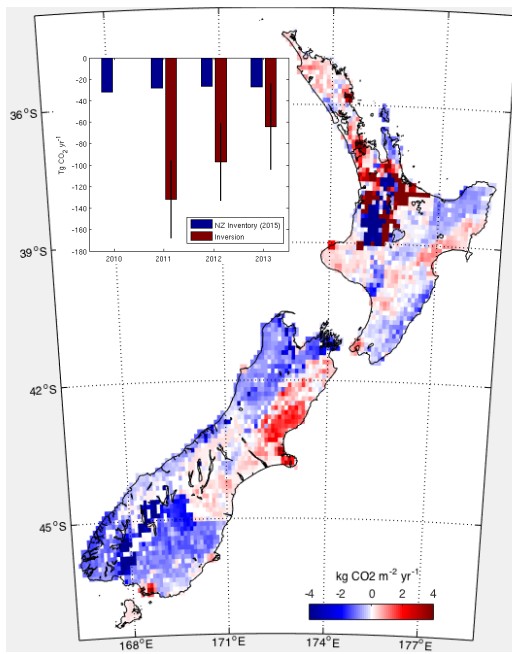

*Figure 11. Geographic distribution of land-to-air CO2 flux, averaged over 2011-2013. Blue and red regions indicate net carbon uptake and release, respectively. Per area ocean fluxes are too small to show on this scale. Fossil fuel emissions are included and reach up to 20 kg $CO_2$ $m^{-2}$ $yr^{-1}$ in a few grid cells (Auckland area). The colour scale is capped to focus on natural fluxes. Inset: Annual mean results compared to the National Greenhouse Gas Inventory Report.*





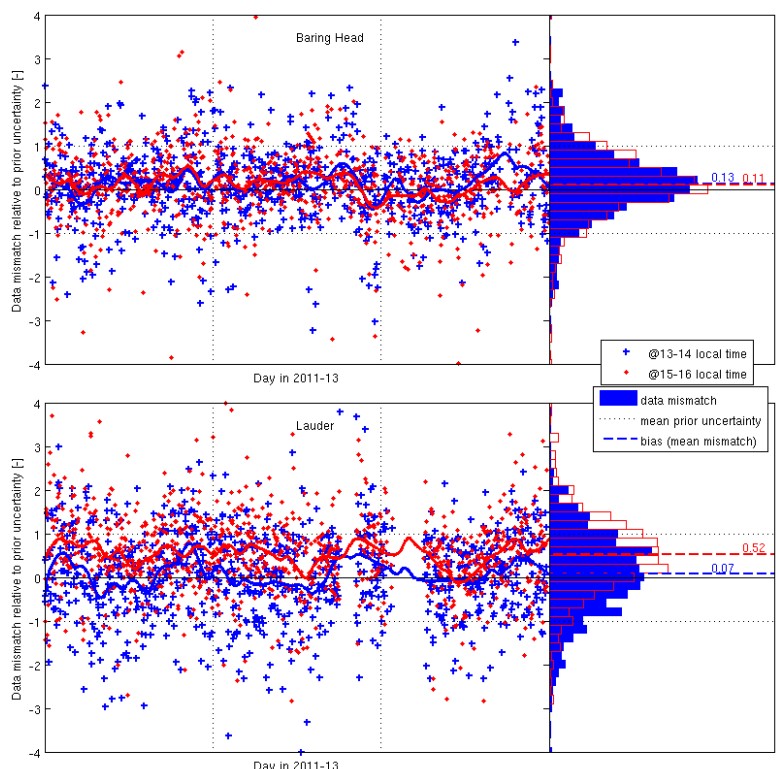

***Figure 12***. *Mismatch (residuals) of modelled vs. observed $CO_2$ in multiples of the prior data uncertainty at Baring Head (top) and Lauder (bottom). Vertical dotted lines separate the years 2011, 2012 and 2013. Solid lines represent a Loess fit with a 3-month window. Horizontal dotted lines mark the prior uncertainty (1σ). The left column shows scatter plots for every day and 1 h release period; the right column shows the mismatch distribution over 2011-2013. Dashed lines with numbers give the bias.*




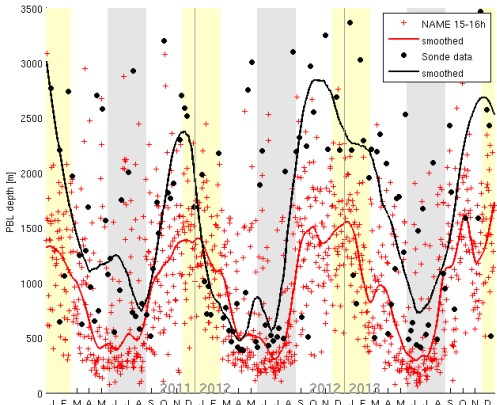

***Figure 13***. *Comparison of boundary layer depth at LAU in NAME at 15:00-16:00 LT and radiosonde observations made at the site (Heffter method). The seasonal cycle has been made more visual using a robust Loess smoother. Summer periods are highlighted in yellow, winter periods in gray.*

