# Peer review of "Atmospheric CO2 observations and models suggest strong"

_Atmospheric Chemistry and Physics, 2016_

## Referee Comment (RC1) · Anonymous Referee #1 · 18 May 2016

This paper presents the first regional CO$_2$ inversion for New Zealand. The inversion yields posterior spatiotemporal CO$_2$-flux distributions which are surprisingly different from the priors (calculated with the Biome-BGC model) and also different from the National Inventory Report.

My overall judgment of the paper is quite favorable, and I recommend publication with minor revisions provided the questions below can be handled satisfactory. A good use has been made of the literature, and the language needs in my opinion but very little correction. The methods are generally well-considered, though some questions remain (see below). Some details, especially the methods to solve the difficult problem of determining the "background" (influence of lateral boundary conditions for the con-

centration) I find difficult to judge from my own experience, but I would trust them from what I have read here.

Descriptions are mostly clear and complete. Some descriptions are quite lengthy, however, and my impression is that especially the descriptions of the observations methods at the Lauder station (Appendix A), and the footprints etc. for each station, could probably be condensed. The discussion about discrepancies between posterior fluxes and inventories seems unusually comprehensive, and of great value. The discussion about the discrepancy between prior and posterior fluxes leads to the important remark that the Biome-BGC model has not been tested for New Zealand conditions, which is a subject which may deserve more emphasis than it gets now in the paper.

*Specific comments*

Page 5, lines 24-27: It is a bit odd that the observation time (with respect to the mean solar time) differs between summer and winter ! To my knowledge, this is unusual. Can it cause a bias in the annual course of the fluxes ?

Page 10, lines 23-25: crops cover a small area, but does this imply that their contribution is small ? They might still contribute at certain times if their flux density is high.

Page 11, line 28: Are the fossil fuel emissions taken time-independent within the years ? Emissions from heating etc. depends on the season.

Page 16, lines 11-12 (and also page 18, lines 16-17): Does this mean that the time-dependency of the fluxes within the week is neglected ? If so, wouldn't this lead to systematic errors (of the kind of rectifier errors or something like that) ? If not, where is the time-dependency described ? This is an important question.

Page 19, line 2: "We excluded the ocean prior": How is this done exactly ? The description is unclear.

[Figure]

Page 19, lines 4-29: This text is so long that it would be logical to make a separate numbered subsection of it.

Page 26, line 1-3: Add more about the background of these findings.

Page 26, lines 21-23: Unclear piece.

Page 27, lines 24-25: Does this hold for all regions ?

Page 28, lines 9-12: I don't understand this. It would seem that with a deeper PBL, the diurnal course in $CO_2$ concentration would be smaller, hence afternoon concentration would be *higher*. Or does neglecting of the diurnal course of the flux play a role here ?

Page 29 end-30 begin: A fuller explanation is found on page 22, line 20, and I suggest to include this also here because it is important for anyone doing research with the biome model. I would also recommend to mention it in the abstract.

Page 44, Table 1: (1) The division into north/south and east/west is not shown in the figures; (2) Only posterior results are given here, it might be interesting to show priors too.

*Small corrections*

Page 17, line 14: instead of "equivalent" I would use "similar" as "equivalent" is a very strong term in mathematics.

Page 25, line 22: add comma after "eddy-covariance".

Page 44, Table 1: "parantheses" should be "parentheses".

Page 50, figure 8: I don't understand the interior legend for the black color (it seems to pertain to Australia).

Page 53, figure 11: "per area fluxes": I would prefer "flux densities" (also page 24, line 2).

---

## Referee Comment (RC2) · Anonymous Referee #2 · 30 May 2016

The paper "Atmospheric CO2 observations and models suggest strong carbon uptake by forests in New Zealand" describes a regional atmospheric inverse modeling framework for the estimate of the weekly to annual mean CO2 land ecosystem fluxes in New Zealand. It evaluates the results using comparisons to estimates from the national inventory.

- General comments

The scope of the manuscript, the atmospheric measurements, the modeling and inversion frameworks as well as the detailed discussion on the comparison between the inversion results and the inventory should, in principle, make this study worth a publication in ACP.

However, there is likely a critical issue regarding the lack of account for the diurnal cycle of the CO2 land natural fluxes both in the prior estimate of these fluxes and during the inversion. The authors need to demonstrate that the impact of this lack of account is not problematic, clarify that they had actually accounted for this if it was the case, or should run new experiments to account for this (see my main comment A)). Furthermore, some analysis are rather weak and the quality of the text critically needs to be improved.

Mainly

A) The actual temporal resolution of the prior estimate of the CO2 land natural fluxes is 1 month. The inversion adjusts 1-week mean fluxes. From my understanding, there is thus no account for the diurnal cycle of these fluxes both in the experimental framework nor in the analysis and discussions of the text. This can be a critical issue given that the system assimilates afternoon data only, and given that, due to the configuration of New Zealand, these data should be primarily influenced by the afternoon, and, to a lesser extent, morning fluxes. By ignoring the diurnal cycle of the CO2 land natural fluxes, the prior would underestimate the natural sink during the afternoon. Consequently, the assimilation of data during the afternoon only would lead to a strong increase of the afternoon sink, which, due to aggregation errors, can be artificially extrapolated into a abnormally high increase of the weekly to annual mean sinks and of the seasonal cycle. Since this is exactly what is observed in the inversion results, this can strongly weaken the confidence in these results. An annual sink of 0.1PgC for NZ is a bit surprising so the authors will definitely need to better support such a number. If the authors actually accounted for this, they should clarify it. If not, they should investigate it and it may be necessary for them to rerun the inversion by separating the adjustments for night-time and daytime fluxes. This problem of the lack of account for the diurnal cycle of the CO2 natural fluxes can be connected to the lack of account for past CO2 regional inverse modeling studies in the introduction and method sections (see my major comment on the introduction: the paper seem to rely on techniques and knowledge from

CH4 inverse modeling experiments, ignoring the potential specificities of the inverse modeling applied to regional CO2 natural flux estimation). In this context, the text at lines 16-20p18 is really embarrassing (in addition to being very confusing).

B) The logic, order and rigor of the text needs to be strongly improved; it would be difficult to detail all the issues corresponding to this comment but I try to list some representative examples below. The order of the figures is not consistent with that of the references to these figures in the text and it often looks like random.

I also think that there is a significant mistake regarding the mathematical framework and configuration of the inversion since I believe that the chi-test that has been used to set up the model errors is wrong. This should have been 2J/n where n is nobs the number of obs and not nobs-nflux (the authors seem to have confused the chi test for J and that for "Jobs" i.e. the part of J corresponding to the misfits to the obs). I assume that this comes from the fact the authors were confused about the type of cost functions analyzed in the ref (Gurney et al 2004 and Baker et al 2006) that they provide for this test. In any case, the way this test is presented is confusing and lacks of rigor.

C) The different focuses of the text could be better balanced. In particular the evaluation of the inversion is a bit short while there is much material on the footprints of the measurements that is not really exploited for such an evaluation. The skill of the transport model for simulating the concentration at the measurements sites is hardly analyzed even though the topography seems quite complex around the sites and the measurements are taken at 10magl only.

D) I feel that there is sometimes a sort of over-confidence in the concept of baseline and in that of constraining the fluxes of different regions using the different measurements sites independently (while a traditional concept of the atmospheric inversion is to exploit gradients between sites to infer fluxes between these sites), and I think that it should be better discussed and weighted when analyzing the results.

The idea of a very slowly varying baseline concentration applying to a very large por-

tion of the model boundaries (the background sectors) may often be unadapted. I do not contest the use of such a concept to solve for the boundary conditions in this study since these conditions are difficult to deal with and this practical solution is among the relevant ones. But the paper could better discuss it and more carefully analyze it in the CO2 data timeseries. CO2 simulations over large domains at high resolution (e.g. https://www.youtube.com/watch?v=x1SgmFa0r04) indicate that synoptic patterns of CO2 can travel over large distances which hamper the concept of baseline. In particular, the influence of CO2 sources and sinks in Australia (not that of the portion included in the domain only) could generate large variations in the concentrations measured in New Zealand at synoptic scales even though it is at more than 2000km from New Zealand. In principle, regional inversion frameworks using dense networks where, for most of the wind directions, parts of the measurement sites are located downwind some of the other sites (with the targeted fluxes located in between) limit the impact of uncertainties in the boundary conditions to the margins of the observation network. However, here, the network seems to have been deliberately set up so that the stations work independently to target different areas of New Zealand (and the text on p3 l. 20 only sees the use of several stations as an "advantage").

The paper considers a baseline uncertainty whose exact computation is quite impossible to understand in the main text (l20-21p13) and in appendix B (B1 does not mention any estimate of uncertainties for the southern baseline, B2 is quite confusing). It is difficult to understand what the authors aim at characterizing with this uncertainty. However, some statistics on the timeseries analyzed for such a computation could help to assess the robustness of the baseline concept. The test of sensitivity to a 0.1ppm bias in the baseline ((i) page 18) does not evaluate the weakness of the concept of baseline that I discuss here as illustrated by the solutions proposed in conclusion for tackling such a bias.

The study could better characterize whether there could be some weeks or month when results could be less robust due to the influence of fluxes from Australia or from

other areas (e.g. by comparing the filtered baseline to the actual measurements when the sensitivity to New Zealand fluxes is relatively low, or when large winds blow from a background sector; and by using the analysis of the footprint and the timeseries of the model-data misfits). All of this should be better introduced when presenting the technique (section 5 is often confusing), and better analyzed and discussed in the result and discussion sections.

- Major comments by section:

1) Abstract

The authors could give some insights on the confidence in the inversion method and estimates; this would help better end this abstract (the present "but some differences are likely to remain" is a bit abrupt)

2) Introduction (1)

At line 30-32, the authors discuss the development and application of regional atmospheric $CO_2$ inversions. However, they cite papers on $CO_2$ footprint modeling and inverse modeling for other species, but no paper on $CO_2$ regional inverse modeling even though there have been a lot studies in this field since more the 5 years. Providing details at the end of page 3 on the papers by Stohl et al. (2009) and Manning et al. (2011), which had to deal with the estimate of sources with very different spatial and temporal patterns compared to the $CO_2$ natural fluxes, is a bit problematic (and one can hardly see the link between these detailed description on page 3 and the specificities of this study on page 4). Therefore, the introduction presents the principle of regional inversion as a sort of generic algorithm that could be applied similarly to any GHG. And it gives a limited view on the range of techniques that have been used for regional atmospheric inversions. This is emphasized by the implicit assumption that the the concept of "baseline" (in the way it is treated in this study) applies to all atmospheric regional inversion cases, while many regional systems, by using outputs from larger scale model to force their boundary conditions and/or by relying on the

spatial gradients between measurement stations to limit their impact, give a different answer to the problem of solving for the influence of fluxes outside the modeling domain. Opposite to what is said at line 2 on page 3, the use of Lagrangian models is not a requirement for regional inversion. In general, the text which attempts at defining the regional inversions vs global inversion from page 2 to page 3 is a bit confusing and could have been more concise.

3) Method sections (2-5)

The logical structure is often confusing. This regularly forces the authors to anticipate for the next sections and thus make redundancies (e.g. the inversion technique is introduced on p8 26-28 and at the beginning of sections 4 and 5.3). Presenting the general frame of the inversion (5.3) could help solve for it.

The mathematical formalism should be based on rigorous notations. Presently, the vector x and matrix T describe completely different things between section 3.2 and section 5.3. The spatial distribution of the fluxes within a region, which is implicitly contained in matrix T in section 5.3, and not in the vector of the prior estimate of the regional budgets x0, are called throughout the text "prior distributions".

The prior uncertainties derived from scientific publications and objective comparisons (section 4.1 and then extrapolated in section 5.3 at the regional scale) seem to be smaller than the artificial "uncertainty component of 50% of the seasonal amplitude" added in section 5.3 (even though it is difficult to understand what l15-17p16 really mean). Therefore, the derivation of the prior uncertainties seems artificial.

The introduction of the term with S in equation (2) raises questions. Why did not the authors rather introduce temporal correlations in the Co matrix to limit the changes in the weekly variations of the fluxes ?

The logic behind the specific choice and configuration of the sensitivity tests on page 18 is not really convincing and does not seem to tackle some of the most critical sources

of errors in the inversion. What is the link between the modeling in NAME (l4p18) and the details given on (ii) later in page 18, which concern the spatial distribution of the fluxes within the regions ? Definitely, a practical assessment of the transport model uncertainties would have been interesting. I do not understand the test of sensitivity to the inclusion of the ocean fluxes in the modeling framework. What does l2p19 mean ? How to connect it to l30p28 ? In principle, the impact of the uncertainty in these fluxes should be correctly accounted for in the reference test. The point maybe be about "biases" but it is difficult to guess, in this section and when analyzing the results, whether the authors systematically make a rigorous use of the term "bias".

4) Result sections (7)

In general, the analysis are a too qualitative.

The robustness of the inversion needs to be better assessed through the analysis of the comparisons between prior and posterior $CO_2$ vs. measurements. I feel that relying on the sensitivity tests to state that the results are robusts (l20-21p1, l8p29) is not really satisfying given that the sensitivity tests do not really sample the most critical sources of errors in the inversion system. The first subsection of section 7 could have been dedicated to such a detailed analysis of the $CO_2$ concentration model - data misfits, providing an opportunity to exploit the long discussions on the station footprints to potentially correlate the highest misfits with specific transport conditions.

In order to ensure that the chi square statistics are right, the authors have to derive a 0.4ppm model error which is surprisingly low for 10magl stations surrounded by a complex topography, and which is strange given that the model data misfits often exceed the projection of the prior uncertainty in the obs space. As said in major comment B, I think that the authors made a wrong chi test but I also assume that this hardly explains such a low diagnostic of the model error. Figure 12 normalize the model data misfits by the prior uncertainty and the text hardly discusses absolute values (in ppm) of these misfits. These absolute values could reveal the skill of the model for simulating $CO_2$ at

the measurement sites. At least, figure 12 reveals large biases (and errors with a high temporal correlation over several months) whose consequences for the confidence in the inversion results should be better weighted.

The confidence in the inversion results at the weekly scale can also be weakened by the quite high week to week variations of the inverted weekly fluxes despite the term with the S matrix in equation 2. However, while the text takes time to discuss this penalty term in section 5.3, it skips the analysis of the week to week variations in the result sections. Does it lower the confidence in the monthly mean results or can it be explained by actual variations or through the variations of the observation constraint ?

One has to trust the authors that their inversion results can be directly compared to a sort of crude NIR total estimate in the abstract and in the introduction. However, the text in 7.3 reveals (even though it is often confusing) many differences between the type of fluxes covered by the inventory and the inversion, and that the inventory provide enough details to filter some major flux components that cannot be accounted for by the inversion, so that a more relevant budget could be derived for this comparison. Therefore, things should be presented differently (i.e. turned the other way around, starting with a presentation of the content of the NIR, and following with an extraction of a relevant budget from the NIR) and, in the abstract,"the sink [derived by the inversion]" should not be compared to "the reported 27 tGco2YR-1" (l. 26-27 p1) but to a more relevant combination of the NIR components.

5) Conclusion (8)

Given all my concerns that are detailed above, I think that the discussion misses the critical needs for the improvement of the inversion in the last paragraphs, and that it is too optimistic regarding the results in the first paragraph and at l1-3p31.

- some questions

* why do not the author assimilate 14:00-15:00 data ? in order to save computations ?

* p6 l3 vs l7: it seems that this site should be strongly influenced by the emissions of Wellington. Can the authors provide more details (e.g. statistics) on this topic than at l23-24p20 ?

* l31p11, l12p2, l12-16p24: if focusing on fossil fuel emissions, what is the difference between EDGAR and NIR ? can we assume that NIR has a far more precise estimate than EDGAR ?

- Sample of minor issues illustrating some of the general comments above

* the text often forgets to be precise about the fluxes that are discussed (l13p1-> natural, l.12 p2: CO2 emissions -> anthropogenic accounting for land use or fossil fuel only ?; first sentence and line 13 of abstract and 1st paragraph page 4: precise that you target natural fluxes; l27p8: the system solves for natural fluxes; l2-3p9 sinks and sources: natural sinks and sources...); on the same topic, lines 12-14 p24: as it is, these sentences do not make sense since land use change emissions are not fossil fuel emissions.

* discussing RBM in section 2 and 5.2 is a bit strange but the authors are embarrassed with the fact that the partitioning of the fluxes accounts for the future inclusion of RBM data in the system for future experiments.

* examples of abusive shortcuts: l13 p1 (from measurement records), l16 p2 (locally present vegetation), p822-24, l7p17 (which quantity does this number corresponds to ? C is a cov matrix)...

* examples of awkward and confusing sentences: l2-3p3, l10-11p3, all sentences from l14 to l21p3, l6p4, l23-24p9 (there is a long discussion on the biomes and land use maps on page 9 and 10 but we hardly understand how they will be used), the first paragraph of 5.1, "monthly standard deviations" l20p13, l4p15,l15-17p16, l20-21p16, l4-6p17, l10p17,l14-15p18, the whole paragraph (ii) on p18, l21p21,12-14 p24, l20p24, l21-23p26, l22-25p27, l26-28p27....

* l18p5: the link between the PBL and the horizontal extent of the footprint of the measurements is a bit confusing

* the logic behind the model representativeness error computation is not obvious (p5l28-32), in principle, the STD of the concentration variability at the 5-min scale does not correspond to the skill of the model to represent hourly measurement averages; the assumption underlying this computation should be explained

* p8 l8-11: the normalization (if I understand it correctly) does not make sense and just loses information.

* the text often forgets to associate numbers with a time or space scale (e.g. l31-32p10, 3rd paragraph of page 11, l7p24...)

* section 6 is poorly connected to the other sections, and the part of section 6 before the start of 6.1 sometimes sounds like a summary of the subsections 6.1 and 6.2

*l14-17p14 and l25-29p19: the diagnostic assesses the sensitivity to the fossil fuel emissions in the part of Australia that is in the modeling domain, not the sensitivity to all fluxes in the whole Australia; in other places, and especially in the conclusion, the paper will state that potential errors from Australia need to be handled.

* the logic behind the justification of the partition of the ocean regions in the second half of p14 is not really clear.

* l5p16: it is not the "uncertainty in the inverse modeling system" (which is something that would be difficult to quantify), it is the likely the uncertainty in the transport model

* l7-10 p23 are poorly connected to the analysis above

* l17-20p27 are wrong, if posterior uncertainties are low, negative correlations between these uncertainties do not mean that the inversion is unable to distinguish the corresponding fluxes; in such cases, it would just mean that the residual uncertainty in the corresponding fluxes is due to such a problem of separation

* labels in the figures are often difficult to read

*l22-24 p21: at this stage of the manuscript, we do not know how the additional uncertainty from the sensitivity tests is included in the figure (this will be explained at l21-25p28 which are quite confusing)

---

## Author Comment (AC1) · 25 Aug 2016

**Revision of ACP manuscript acp-2016-254**

**"*Atmospheric CO$_2$ observations and models suggest strong carbon uptake by forests in New Zealand*"**

by K. Steinkamp, S.E. Mikaloff Fletcher, G. Brailsford, D. Smale, S. Moore, E.D. Keller, W.T. Baisden, H. Mukai and B.B. Stephens

August 20$^{th}$ 2016

Dear Editors,

The co-authors and I would like to thank the anonymous referees #1 and #2 for providing constructive reviews. They helped us to improve and clarify the paper. We have responded fully to both reviews below, and we would be delighted to submit a revised manuscript.

This process has taken a bit longer than desired, largely owing to our decision to develop an additional test case for our model to address concerns of referee #2 regarding the diurnal cycle of CO$_2$ land fluxes in New Zealand.

We thank you for your continued consideration of this manuscript.

Sincerely,

Kay Steinkamp

**Anonymous Referee #1 – Interactive Comment**

*This paper presents the first regional CO₂ inversion for New Zealand. The inversion yields posterior spatiotemporal CO₂-flux distributions which are surprisingly different from the priors (calculated with the Biome-BGC model) and also different from the National Inventory Report.*

*My overall judgment of the paper is quite favorable, and I recommend publication with minor revisions provided the questions below can be handled satisfactory. A good use has been made of the literature, and the language needs in my opinion but very little correction. The methods are generally well-considered, though some questions remain (see below). Some details, especially the methods to solve the difficult problem of determining the "background" (influence of lateral boundary conditions for the con-centration) I find difficult to judge from my own experience, but I would trust them from what I have read here.*

We thank the referee for his/her support and the helpful and insightful comments. We believe we have been able to improve this manuscript by implementing many of this referee's comments, although not all, as explained in detail below.

*Descriptions are mostly clear and complete. Some descriptions are quite lengthy, however, and my impression is that especially the descriptions of the observations methods at the Lauder station (Appendix A), and the footprints etc. for each station, could probably be condensed.*

We agree that the description of the Lauder station in Appendix A is unusually lengthy. In order to address this, we have condensed sections A.1, A.3 and A.4 and reduced the overall section size by about half a page (the new section is printed below). Usually, we would reference to a published paper that is dedicated to explaining the Lauder site and the instruments installed and operated there. Then it would be possible to provide a summarized description here, similar to our handling of the Baring Head station, instruments and data, which are described in detail by, e.g., Brailsford et al (2012) and Stephens et al. (2013). However, the Lauder CO₂ dataset is a new dataset introduced to the community and, to date, there is no published source available describing the site. Thus, it is essential to provide a conscientious, complete and transparent description here. We put the detailed descriptions in an appendix, so as to not disrupt the flow of the manuscript, while keeping a brief summary in the main text. This also allows for Appendix A to be used in future papers as the site/instrument reference.

Similarly, the station footprint analyses contain a wealth of information that is very relevant to this work and future work but have not been published elsewhere.

[revised manuscript text omitted]

*The discussion about discrepancies between posterior fluxes and inventories seems unusually comprehensive, and of great value.*

Thank you for this comment; it is much appreciated.

*The discussion about the discrepancy between prior and posterior fluxes leads to the important remark that the Biome-BGC model has not been tested for New Zealand conditions, which is a subject which may deserve more emphasis than it gets now in the paper.*

We would like to clarify that although the Biome-BGC model has not been tested specifically for net ecosystem exchange in New Zealand, the model parameterization used here for managed grassland (sheep/beef and dairy pasture systems) has been calibrated for pasture production (or net primary productivity) under New Zealand conditions as described in section 4.1 (pages 9-11) and references therein. Keller et al. (2014) detail the method used to optimize model parameters using pasture growth data from 6 sites in New Zealand (3 sheep/beef and 3 dairy) as well as validation of the parameterization at an additional 22 sites (sheep/beef) and a comparison of model output to national milk production data (dairy). In addition, as mentioned at the top of page 11, modelled live stem carbon for the default parameterization of the evergreen needle-leaf forest (ENF) biome has been checked against the New Zealand National Exotic Forest Description yield tables (*pinus radiata*; MPI 2012) and is in good general agreement at a regional level.

Several biomes – most notably, evergreen broadleaf forest or EBF – are indeed untested for New Zealand conditions. In contrast to managed grasslands and planted forests, these biomes are largely undisturbed primary or minimally disturbed secondary forests, and we judged that they were unlikely to diverge substantially from default parameterisations that have been tested globally, but not specifically for New Zealand ecosystems. This could add a significant amount of uncertainty to the priors. Given this, it is perhaps not surprising that the prior and posteriors differ greatly in areas where indigenous broadleaf forest dominates, such as Fiordland (noted on page 22 lines 18-22). We have added some text in section 4.1 to emphasize this fact:

Page 10 line 30 – page 11 line 6:
An uncertainty estimate is computed for the a priori $CO_2$ flux from each grid cell. Based on Keller et al. (2014) and personal communication with the authors, we assign a 10% uncertainty for pasture fluxes. For forests, we assign 10% everywhere except in the Canterbury and Otago regions in the South Island, where 56% and 36% are used, respectively. These are conservative estimates based on a comparison of the Biome-BGC ENF modelled live stem carbon with the national exotic forest regional yield tables (MPI, 2012). The Canterbury and Otago regions were assigned larger uncertainties to reflect the larger discrepancy between the Biome-BGC model and the yields in these regions. The uncertainty is taken into account by the Bayesian optimization (Section 5). We note that the EBF biome has not been calibrated or tested under New Zealand conditions and might contain additional uncertainty.

*Specific comments*

*Page 5, lines 24-27: It is a bit odd that the observation time (with respect to the mean*

*solar time) differs between summer and winter ! To my knowledge, this is unusual. Can it cause a bias in the annual course of the fluxes ?*

The choice to align the observation times with summer/winter time in New Zealand was based on the observation that gas inlets at varying heights at Lauder showed the least deviations from one another during the 15:00-16:00 local time window. This is indicative of a well-mixed atmospheric boundary layer that can be confidently modelled (the NZLAM model can not properly resolve vertical processes at the meter-scale corresponding to the height difference of inlets). Later on, as we additionally incorporated observations from the 13:00-14:00 time window, this reasoning became less important. Conditions are generally a bit less well-mixed at those times, and a one hour shift had little influence. In order to have a consistent setup for both time slots, we decided to stick to the local time windows.

In principle, this could cause shifts in the simulated fluxes around the transition between summer and winter times, however, such shifts are not visible in the results. In addition, a comparison for both time slots reveals that, for almost every day, the footprints are indistinguishable by visual inspection. Since the time slots are two hours apart, it appears very unlikely that a shift of one hour could introduce any significant bias.

*Page 10, lines 23-25: crops cover a small area, but does this imply that their contribution is small ? They might still contribute at certain times if their flux density is high.*

It is true that cropland could contribute over-proportionally at certain times. For all but one region (South Island east, region index 12) the fraction of cropland area to total area is less than 1.7%, so that the cropland per-area flux density would have to be unrealistically large to be comparable to the contributions of the other biomes. In region 12, the area fraction is 8.7%, making it more likely for the inversion to be influenced by cropland fluxes. The prior flux is not a strong constraint in that region (prior uncertainty is around 100 Mt $CO_2$ $yr^{-1}$ for the weekly fluxes) and the area is well sampled by the observing network (Figure 9). Thus, a strong signal from cropland in region 12 is likely to be accurately detected by the inverse method without substantial bias from the prior uncertainty.

*Page 11, line 28: Are the fossil fuel emissions taken time-independent within the years ? Emissions from heating etc. depends on the season.*

Yes, the fossil fuel emissions are constant throughout the 2011-2013 period. As we describe in section 4.3, we used the annual emissions available from the EDGAR v4.2 dataset to extrapolate emissions for the years 2011-2013. While the EDGAR dataset is well suited to be used in a study like ours, it does not include intra-annual variations in emissions.

The influence of fossil fuel emissions is small at Baring Head and negligible at Lauder relative to the contribution from the terrestrial biosphere (Figure 7). This is because New Zealand is a sparsely populated country, with a population density of 15 people per square kilometre. Auckland, New Zealand's only major city and home to over a third of New Zealand's 4.2 million people, is not well sampled by our current observing network (Figure 9). Thus we consider it unlikely that the inverse estimates are substantially influenced by the lack of seasonal cycle in the fossil fuel emissions.

*Page 16, lines 11-12 (and also page 18, lines 16-17): Does this mean that the time-dependency of the fluxes within the week is neglected ? If so, wouldn't this lead to systematic errors (of the kind of rectifier errors or something like that) ? If not, where is the time-dependency described ? This is an important question.*

Yes, both the terrestrial and oceanic fluxes are based on flux maps from the Biome-BGC and Takahashi datasets that are available at monthly resolution (described in sections 4.1 and 4.2). They are interpolated to weekly maps, which are then integrated over the region areas and used as the weekly prior flux estimates. The combination of weekly flux resolution for 25 regions with twice-daily observations from two stations ensures that the Bayesian computation is sufficiently constrained by the observations, i.e. there are more observations than unknowns. With our approach, it is not feasible to estimate regional fluxes at daily, or even hourly, resolution.

While not of concern for oceanic fluxes, for terrestrial fluxes this could lead to biases due to the strong diurnal cycle exhibited by the terrestrial biosphere. These biases should be strongest near the stations where their sensitivity is greatest. The region layout described in section 5.2 includes small regions around each station to contain all local effects that could otherwise be attributed to larger regions further upwind – this was partly done with the diurnal cycle in mind. We also include a sensitivity case (section 5.4) where we change the prescribed within-region flux pattern, because the inversion's interpretation of a sudden flux peak not resolvable at the weekly scale is sensitive to these patterns. As reported in the paper, we did not find a bias in that sensitivity case strong enough to interfere with the conclusions.

However, in response to this comment and to the main comment of referee #2, we conducted an additional synthetic experiment where we explicitly compare the potential bias arising from the fact that the terrestrial diurnal cycle is not resolved. We report the details as part of our response to referee #2's comment.

*Page 19, line 2: "We excluded the ocean prior": How is this done exactly ? The description is unclear.*

This is done by applying an uncertainty of $10^8$ Tg $CO_2$ yr$^{-1}$ to the regional prior fluxes, such that the prior estimate has a negligible effect on the inverse estimates. To choose this value, we first ramped up prior uncertainties until no further changes in the posterior estimation were obtained (around 1000 Tg $CO_2$ yr$^{-1}$), then added 5 more orders of magnitude. We did not choose even higher values to avoid losing computational accuracy when dividing by those values during the inversion procedure.

We have added some text in section 5.4 to clarify:

(iii) Estimates of terrestrial $CO_2$ fluxes in New Zealand are influenced by the ocean flux prior through atmospheric transport. After entering the model domain at baseline levels, the air travels inevitably over a large stretch of ocean and will arrive at the New Zealand coast carrying an oceanic signal in its $CO_2$ concentration. Errors in the *a posteriori* ocean flux estimates will be interpreted by the inversion as terrestrial $CO_2$ flux. In a sensitivity test, we excluded the ocean prior to isolate its impact on the results. In order to accomplish this, the oceanic prior uncertainty was raised to $10^8$ Tg $CO_2$ yr$^{-1}$.

*Page 19, lines 4-29: This text is so long that it would be logical to make a separate numbered subsection of it.*

The intended structure of section 6 is such that we give a concise overview of the outcomes of the analyses, before diving into a bit more detail in the site-specific subsections 6.1 and 6.2. We clarified this in the text:

Generally, $CO_2$ measurements at BHD are most sensitive to sinks and sources in the Southern Ocean (south of 55°S), the Tasman Sea and the South Island. Australia and the North Island influence BHD $CO_2$ to a lesser extent. Observations at LAU are strongly influenced by local to regional terrestrial sinks and sources of $CO_2$, enabling the station to see air from a large portion of the southern South Island. Further site-specific details are given in the subsections below.

*Page 26, line 1-3: Add more about the background of these findings.*

We cite recent studies finding a large difference in soil carbon cycling on hilly vs. flat pastures to help explain part of the reason that the NIR estimates differ from our study. They are also used as an example to highlight the large uncertainties that exist due to up-scaling estimates based on a limited number of of sampling sites. Since this is only an example (we do not use their values in any computation), and keeping in mind that section 7.3 is already rather lengthy, we find it preferable not to elaborate further here. We have emphasized this in the text:

For example, a recent analysis of all sites available nationally suggests that sites on flat pasture are losing soil carbon at rates of ~170 g $CO_2$ m$^{-2}$ yr$^{-1}$, while sites in hill country are gaining ~770 g $CO_2$ m$^{-2}$ yr$^{-1}$ (Schipper et al., 2014).

*Page 26, lines 21-23: Unclear piece.*

The intention of the text in question was to summarize some of the discussions in section 7.3. Due to being unclear and also repetitive, we have removed it.

*Page 27, lines 24-25: Does this hold for all regions ?*
Yes it does, all regional error correlations were included in the analysis. We have clarified this in the text:

An analysis of all regional error correlations reveals that both negative and positive correlations are present, however, only 0.13% of all pairwise correlations have an absolute value greater than 0.1. Very few values are smaller than -0.4 or greater than 0.2, with the negative extreme around -0.7 and the positive extreme around 0.3. Hence, with the available data, the inversion appears able to resolve weekly fluxes at the regional level chosen.

*Page 28, lines 9-12: I don't understand this. It would seem that with a deeper PBL, the diurnal course in CO2 concentration would be smaller, hence afternoon concentration would be higher. Or does neglecting of the diurnal course of the flux play a role here ?*

We agree that the description was not very clear, as potential difficulties of modelling PBL depth during summer afternoons couples to the diurnal flux cycle, which is not resolved in our study. We discuss this in more detail as part of our response to referee #2's comment concerning the diurnal cycle.

*Page 29 end-30 begin: A fuller explanation is found on page 22, line 20, and I suggest to include this also here because it is important for anyone doing research with the biome model. I would also recommend to mention it in the abstract.*

We have added text in the conclusions (page 30) to emphasize that we are comparing our estimates to Biome-BGC's EBF category:

Regions covered predominantly by indigenous forest appear to have more pronounced photosynthetic and respiratory activity than suggested by the land model's evergreen broadleaf forest (EBF) category.

To keep the abstract concise, we abstained from adding the information there also.

*Page 44, Table 1: (1) The division into north/south and east/west is not shown in the figures; (2) Only posterior results are given here, it might be interesting to show priors too.*

In Fig 10, the region indices making up each aggregate region are given in parentheses underneath the aggregated region's name. Together with Fig 6, this should give detailed information about the aggregated regional outlines. We clarified this in the caption of Fig 10:

**Figure 10**. *Weekly $CO_2$ fluxes in 2011-2013 from selected regions, in Tg $CO_2$ $yr^{-1}$. Prior flux estimates are shown in gray, and the inversion results are shown in blue. Shaded areas represent flux uncertainty (1s). The cyan shade represents the extra uncertainty obtained from the sensitivity cases. Note that there is a one-off change in scale of the flux axis for sub-island scale regions. A positive flux indicates a net release of $CO_2$ to the atmosphere, while a negative value indicates uptake by the land biosphere. Regional indices in parentheses correspond to Figure 6.*

*Small corrections*

*Page 17, line 14: instead of "equivalent" I would use "similar" as "equivalent" is a very*

*strong term in mathematics.*
*Page 25, line 22: add comma after "eddy-covariance".*
*Page 44, Table 1: "parantheses" should be "parentheses".*

Thank you. We have made the changes above.

*Page 50, figure 8: I don't understand the interior legend for the black color (it seems to pertain to Australia).*

We apologize for using a rather informal legend for Australia, and have replaced it with "AUS".

*Page 53, figure 11: "per area fluxes": I would prefer "flux densities" (also page 24, line 2).*

While perhaps less elegant, we prefer to keep the current phrasing for it not to be confused with densities in a 3-dimensional volume.

**Anonymous Referee #2 – Interactive Comment**

*The paper "Atmospheric CO2 observations and models suggest strong carbon uptake by forests in New Zealand" describes a regional atmospheric inverse modeling framework for the estimate of the weekly to annual mean CO2 land ecosystem fluxes in New Zealand. It evaluates the results using comparisons to estimates from the national inventory.*

*- General comments*

*The scope of the manuscript, the atmospheric measurements, the modeling and inversion frameworks as well as the detailed discussion on the comparison between the inversion results and the inventory should, in principle, make this study worth a publication in ACP.*
*However, there is likely a critical issue regarding the lack of account for the diurnal cycle of the CO2 land natural fluxes both in the prior estimate of these fluxes and during the inversion. The authors need to demonstrate that the impact of this lack of account is not problematic, clarify that they had actually accounted for this if it was the case, or should run new experiments to account for this (see my main comment A)). Furthermore, some analysis are rather weak and the quality of the text critically needs to be improved.*

We appreciate the referee's feedback about biases that might be introduced by our model's inability to resolve the diurnal cycle of the $CO_2$ land natural fluxes.

In order to address the referee's concerns, we designed and ran new experiments. Hourly $CO_2$ land fluxes were estimated based on a simple reconstruction of diurnally-varying NEP from Biome-BGC and combined with the NAME model to construct a synthetic data set that includes diurnal variability in the fluxes. Then, we used our current inversion framework, which neglects diurnal variability, to estimate the fluxes from this synthetic dataset. Biases from the diurnal cycle will then lead to differences between the known fluxes used to generate the synthetic data and the *a posteriori* flux estimates.

For the regions in the South Island, the results indicate that the lack of account for a diurnal cycle in the control run is not problematic in the context of the conclusions of this study. Weekly

and annual-mean biases do occur in some regions, but they are generally smaller than our posterior uncertainty estimates. Further, this experiment does not reveal any systematic biases towards more uptake across all or land regions.

Details follow in the context of main comment A) and will be incorporated into the manuscript.

We have worked to clarify and/or strengthen aspects of the analysis, wherever the referee suggested as part of their specific comments below. Similarly, we sought to improve the quality of the text in places specifically pointed out by the referee.

*Mainly*
*A) The actual temporal resolution of the prior estimate of the CO2 land natural fluxes is 1 month. The inversion adjusts 1-week mean fluxes. From my understanding, there is thus no account for the diurnal cycle of these fluxes both in the experimental framework nor in the analysis and discussions of the text. This can be a critical issue given that the system assimilates afternoon data only, and given that, due to the configuration of New Zealand, these data should be primarily influenced by the afternoon, and, to a lesser extent, morning fluxes. By ignoring the diurnal cycle of the CO2 land natural fluxes, the prior would underestimate the natural sink during the afternoon. Consequently, the assimilation of data during the afternoon only would lead to a strong increase of the afternoon sink, which, due to aggregation errors, can be artificially extrapolated into a abnormally high increase of the weekly to annual mean sinks and of the seasonal cycle. Since this is exactly what is observed in the inversion results, this can strongly weaken the confidence in these results. An annual sink of 0.1PgC for NZ is a bit surprising so the authors will definitely need to better support such a number. If the authors actually accounted for this, they should clarify it. If not, they should investigate it and it may be necessary for them to rerun the inversion by separating the adjustments for nighttime and daytime fluxes.*

We agree that the lack of a diurnal cycle within the inversion framework could lead to biases as described by the referee. In our original submission, we had already made an attempt to minimize this issue by explicitly separating small local regions around each station. Any potential bias caused by an increased afternoon sink in summer should be most pronounced near the stations, because the further away we go, the more the flux signals are integrated across hours and days. This bias of increased weekly sinks in summer should therefore be most prominent in those local regions, but is in fact not observed there. Furthermore, the reference inversion does not only show an enhanced summer sink compared to the prior but also enhanced emissions in winter, when the diurnal effect is much weaker.

However, we acknowledge the importance of this comment and have investigated the issue more thoroughly by designing a synthetic data experiment to explicitly compare inversion results to a synthetic "truth" that contains a diurnal cycle. To this end, we prepared hourly prior fluxes by imposing a simplified diurnal cycle on daily, gridded Biome-BGC outputs of GPP, NPP and HR. The diurnal variation in GPP is based on hourly solar insolation, and HR is assumed to occur at a constant rate throughout the day. While this greatly simplifies and exaggerates actual diurnal variation (e.g., it lacks factors that decrease the amplitude of the cycle, such as temperature dependence of respiration or down-regulation of photosynthesis under hot/dry conditions), for the purpose of our experiment it is sufficient to reveal model biases. The analysis covers one year, 2012. We then conducted additional NAME runs over the year 2012 and stored hourly footprints for BHD and LAU for the 15:00-16:00 LT time slot (the most relevant for the afternoon sink issue). These could then be used to translate the hourly prior into synthetic data for the 15:00-16:00 LT time slot. These data were used in an inversion with largely the same setup (data uncertainty, ocean prior, land prior was adapted – see details below) as the reference inversion and weekly posterior flux estimates compared to the true fluxes, i.e. the weekly averages of the hourly prior flux from Biome-BGC.

The figure below, which we included in the revised manuscript, shows the deviation of the posterior estimates from the truth (posterior minus true fluxes). In the revised manuscript, we discuss the results and conclude that the lack of a diurnal cycle is not problematic in most regions, unless they are already severely under-constrained (northern North Island). While in some regions the expected bias of increased weekly or annual sinks in summer is visible, it is contained in the uncertainty envelope. In other regions, and surprisingly this includes the two local regions where the diurnal bias should be most pronounced, we actually find a slightly

suppressed sink along with no discernible seasonal pattern. We added this discussion with more details to the "uncertainties and biases" section 7.4 in the context of the observed model-data mismatch for the 15:00-16:00 time slot, where we also discuss possible issues caused by misrepresented boundary layer dynamics:

[revised manuscript text omitted]

*This problem of the lack of account for the diurnal cycle of*
*the CO2 natural fluxes can be connected to the lack of account for past CO2 regional*
*inverse modeling studies in the introduction and method sections (see my major comment*
*on the introduction: the paper seem to rely on techniques and knowledge from*
*CH4 inverse modeling experiments, ignoring the potential specificities of the inverse*
*modeling applied to regional CO2 natural flux estimation). In this context, the text at*
*lines 16-20p18 is really embarrassing (in addition to being very confusing).*

We would like to clarify that the inclusion of studies concerning other greenhouse gases in the introduction and method sections is due to similarities in the models used, i.e., NAME and/or regional versions of the UK MetOffice's UM. Data handling, baseline analysis and inversion specifics are quite different from those studies, as pointed out in the manuscript.

We have added references to additional studies dedicated to regional $CO_2$ inversions in the introduction of our revised manuscript, to address this point and another one below. However,

most regional $CO_2$ inversions have been developed for the Northern Hemisphere, mostly covering Europe or North America. Characteristics such as dense observing networks allow for a different treatment of key aspects of the inversion (e.g., the treatment of background $CO_2$), which are not feasible in New Zealand, lessening the similarities to those studies somewhat.

We have edited the text at p18 l16-20 so that a generic flux pulse is no longer discussed (even though the diurnal cycle can be described as a series of flux pulses). Rather, the focus is now on the within-region flux pattern, while the discussion of the diurnal cycle has been moved into section 7.4 in response to comment A).

The geographic distribution of the $CO_2$ fluxes is fixed within each region. This can lead to biases in the estimated flux if the region is being unevenly sampled. That is, if a specific observation is sensitive to only a small area inside the region, then the flux estimate for the entire region will be biased towards that area, which may not be representative for the region.

*B) The logic, order and rigor of the text needs to be strongly improved; it would be difficult to detail all the issues corresponding to this comment but I try to list some representative examples below. The order of the figures is not consistent with that of the references to these figures in the text and it often looks like random.*
*I also think that there is a significant mistake regarding the mathematical framework and configuration of the inversion since I believe that the chi-test that has been used to set up the model errors is wrong. This should have been 2J/n where n is nobs the number of obs and not nobs-nflux (the authors seem to have confused the chi test for J and that for "Jobs" i.e. the part of J corresponding to the misfits to the obs). I assume that this comes from the fact the authors were confused about the type of cost functions analyzed in the ref (Gurney et al 2004 and Baker et al 2006) that they provide for this test. In any case, the way this test is presented is confusing and lacks of rigor.*

Thank you for catching the inconsistent ordering of the figures (which was due to changes being made to the references to some figures without updating their order of appearance). We have changed the order of the figures such that they are now in line with their references in the text.

We agree with the referee on the chi-squared test. Our intention was to write n=nobs+nflux-nflux, to reflect the test is indeed for J and not for Jobs. Unfortunately, the +nflux term was forgotten. However, the computation of the test has always been correct in the inversion scripts. We corrected the description in the text in section 5.3.

The reduced chi-squared statistic $\chi^2=2J/n$   is used to assess the fit of the inverse model to the observations (other examples of how this statistic can be used are described in, e.g., Gurney et al., 2004; Baker et al., 2006), where n is the number of observations.

The references mentioned were chosen because they are some of the few that explicitly mention that they have done a reduced chi-squared test, not because we have simply copied their formulas.

*C) The different focuses of the text could be better balanced. In particular the evaluation of the inversion is a bit short while there is much material on the footprints of the measurements that is not really exploited for such an evaluation. The skill of the transport model for simulating the concentration at the measurements sites is hardly analyzed even though the topography seems quite complex around the sites and the measurements are taken at 10magl only.*

We have worked to condense appendix A as part of our response to referee #1. As noted in our response to that referee, much of this material has not been previously reported elsewhere, and therefore it is essential to provide many of the details in order for readers to evaluate the work.

While we use only measurements from the 10m inlets, we did comparisons with data from other heights (e.g., 6m at Lauder) during periods when both are available. For the time slots in this paper (13:00-14:00 and especially 15:00-16:00) there are only small differences, which led to their selection, as described in section 2. While a more in depth analysis of transport model

bias is outside the scope of this paper, recent modelling developments will allow us to explore the impact of using a higher resolution (1.5 km$^2$) version of the model in the near future. A detailed study of the impact of the transport model with a focus on topography will be the subject of a future manuscript.

*D) I feel that there is sometimes a sort of over-confidence in the concept of baseline and in that of constraining the fluxes of different regions using the different measurements sites independently (while a traditional concept of the atmospheric inversion is to exploit gradients between sites to infer fluxes between these sites), and I think that it should be better discussed and weighted when analyzing the results.*
*The idea of a very slowly varying baseline concentration applying to a very large portion of the model boundaries (the background sectors) may often be unadapted. I do not contest the use of such a concept to solve for the boundary conditions in this study since these conditions are difficult to deal with and this practical solution is among the relevant ones. But the paper could better discuss it and more carefully analyze it in the CO2 data timeseries. CO2 simulations over large domains at high resolution (e.g. https://www.youtube.com/watch?v=x1SgmFa0r04) indicate that synoptic patterns of CO2 can travel over large distances which hamper the concept of baseline. In particular, the influence of CO2 sources and sinks in Australia (not that of the portion included in the domain only) could generate large variations in the concentrations measured in New Zealand at synoptic scales even though it is at more than 2000km from New Zealand. In principle, regional inversion frameworks using dense networks where, for most of the wind directions, parts of the measurement sites are located downwind some of the other sites (with the targeted fluxes located in between) limit the impact of uncertainties in the boundary conditions to the margins of the observation network. However, here, the network seems to have been deliberately set up so that the stations work independently to target different areas of New Zealand (and the text on p3 l. 20 only sees the use of several stations as an "advantage").*

At the time of our study, there were two stations in New Zealand where $CO_2$ concentrations are measured *in situ*: Baring Head and Lauder. As described in section 2.1 and 2.2, the data retrieval and processing procedures are very similar between the stations and regular, detailed comparisons are undertaken to ensure that measurements are comparable (as detailed in Brailsford, 2012). In this context, our "setup of the observational network" simply means that we have selected every station that can contribute to inform an inversion for our region.

We agree that a denser network of stations would be advantageous, and we are working to expand our national observing network. However, previous regional inversion studies have demonstrated that it is still possible with a single station to develop a regional inversion, as has been done, for example, for Mace Head, to put constraints on GHG budgets for the whole of Great Britain (e.g., Manning et al., 2011).

We have added text to section 5.1 to clarify some caveats about the baseline uncertainty:

A caveat of this concept of baseline is that synoptic patterns of $CO_2$ can travel over large distances and cause variations in the background concentrations at the regional boundary, which are not accounted for by the baseline formed here. Given New Zealand's geographic isolation and location in the Southern Hemisphere, the strength of synoptic variations is limited compared to regions in, e.g., Europe or North America. While an explicit treatment of synoptic variability is beyond the scope of this study, a sensitivity experiment is described in section 5.4 to address potential biases in the baseline.

Due to the configuration of New Zealand and the orientation of these stations with the average south-westerly winds, Baring Head is located downwind of Lauder for about 25% of the time (based on the southwesterly cluster in Figure 4) and Lauder is practically never downwind of Baring Head. It is therefore difficult to base an inversion on inter-station $CO_2$ gradients only, or to mitigate the influence of synoptic variations along the boundary. When we set up the inversion, we considered two ways to describe $CO_2$ concentrations and variations along the boundary: using modelled $CO_2$ based on global inversions, or constructing a baseline. We selected the second option to facilitate a more data-driven approach and because the southern baseline based on the clean air sector at Baring Head was already well studied and established. The northern component of our baseline is less well established, but also plays a significantly deweighted role in the construction of the weighted baseline. In this context, our

sensitivity test for a baseline bias is to ensure some degree of robustness of estimated fluxes. Of course, while driven more directly by observations, our baseline cannot reflect synoptic influences accurately. However, while synoptic patterns can travel large distances, their strength is much reduced between Australia and New Zealand when compared to the Northern Hemisphere, e.g., the North Atlantic, as is also evident in the video link provided by the referee.

*The paper considers a baseline uncertainty whose exact computation is quite impossible to understand in the main text (l20-21p13) and in appendix B (B1 does not mention any estimate of uncertainties for the southern baseline, B2 is quite confusing). It is difficult to understand what the authors aim at characterizing with this uncertainty. However, some statistics on the timeseries analyzed for such a computation could help to assess the robustness of the baseline concept. The test of sensitivity to a 0.1ppm bias in the baseline ((i) page 18) does not evaluate the weakness of the concept of baseline that I discuss here as illustrated by the solutions proposed in conclusion for tackling such a bias.*

We have added text in B.1 to clarify the computation of the uncertainty of the southern baseline component:

The remainder time series, which is the difference between the monthly record and the sum of the seasonal and trend components from the STL analysis, is used as uncertainty estimate for the southern baseline.

This should also make the computation of the uncertainty of the southern baseline component more clear in B.2.

*The study could better characterize whether there could be some weeks or month when results could be less robust due to the influence of fluxes from Australia or from other areas (e.g. by comparing the filtered baseline to the actual measurements when the sensitivity to New Zealand fluxes is relatively low, or when large winds blow from a background sector; and by using the analysis of the footprint and the timeseries of the model-data misfits). All of this should be better introduced when presenting the technique (section 5 is often confusing), and better analyzed and discussed in the result and discussion sections.*

The difference between the filtered southern baseline and the actual measurements during periods when the conditions for clean air are fulfilled is actually a direct part of the uncertainty of the baseline timeseries and as such directly influences the weight of the associated observations in the inversion. This was probably not described very clearly, but we believe it has become clear with the additions to appendix B discussed above.

We conducted an analysis to look specifically at the frequency of events when Australian fluxes can influence measured $CO_2$. The analysis is described in section 5.2 and further discussed in section 6. Those events are rare, with a frequency of a few days (less than 10) per year. The analysis is based on fossil emissions and as such limited, but it gives an idea about the relative importance of Australian emissions versus emissions from New Zealand. Fossil emissions from New Zealand are much weaker than from Australia in absolute terms, but exert a stronger and much more continuous influence on the measurements, as seen in Figure 7.

*- Major comments by section:*

*1) Abstract*
*The authors could give some insights on the confidence in the inversion method and estimates; this would help better end this abstract (the present "but some differences are likely to remain" is a bit abrupt)*

We have added text at the end of the abstract to point out the main sources of error:

Baseline uncertainy, model transport uncertainty and limited sensitivity to the northern half of the North Island are the main contributors to flux uncertainty.

2) Introduction (1)
At line 30-32, the authors discuss the development and application of regional atmospheric
CO2 inversions. However, they cite papers on CO2 footprint modeling and
inverse modeling for other species, but no paper on CO2 regional inverse modeling
even though there have been a lot studies in this field since more the 5 years. Providing
details at the end of page 3 on the papers by Stohl et al. (2009) and Manning et al.
(2011), which had to deal with the estimate of sources with very different spatial and
temporal patterns compared to the CO2 natural fluxes, is a bit problematic (and one
can hardly see the link between these detailed description on page 3 and the specificities
of this study on page 4). Therefore, the introduction presents the principle of
regional inversion as a sort of generic algorithm that could be applied similarly to any
GHG. And it gives a limited view on the range of techniques that have been used for
regional atmospheric inversions. This is emphasized by the implicit assumption that
the the concept of "baseline" (in the way it is treated in this study) applies to all atmospheric
regional inversion cases, while many regional systems, by using outputs
from larger scale model to force their boundary conditions and/or by relying on the
spatial gradients between measurement stations to limit their impact, give a different
answer to the problem of solving for the influence of fluxes outside the modeling domain.
Opposite to what is said at line 2 on page 3, the use of Lagrangian models is
not a requirement for regional inversion. In general, the text which attempts at defining
the regional inversions vs global inversion from page 2 to page 3 is a bit confusing and
could have been more concise.

We have edited parts of the introduction to rebalance the studies discussing regional inversions of greenhouse gases other than $CO_2$ with those for $CO_2$. In particluar, we have added three references dedicated to regional CO2 inversions: Gerbig et al. (2003), Matross et al. (2006) and Lauvaux et al. (2008).

To address those scales, regional atmospheric greenhouse gas inversions  (Lin et al., 2003; Stohl et al., 2009; Bergamaschi et al., 2010; Manning et al., 2011), including  $CO_2$ inversions (Gerbig et al., 2003; Matross et al., 2006; Lauvaux et al., 2008), have been developed and used to estimate the carbon budgets of regions like Europe and the USA as well as individual nations. A regional inversion can provide top-down $CO_2$ exchange estimates from atmospheric $CO_2$ measurements and transport model simulations that describe the source or sink regions influencing each measurement. They are complementary to bottom-up inventories and provide a means to verify national inventories.

In their inversion study, Stohl et al. (2009) estimate emissions for three HFC and HCFC greenhouse gases on national to global scales for 2005-2007. Their approach uses the FLEXPART Lagrangian model to describe the recent air history arriving at nine observation stations distributed globally. They use a priori emission maps and estimate both the baseline and the regional emissions as part of the inverse modeling. Manning et al. (2011) use 20 years of in situ $CH_4$ and $N_2O$ observations from a single station, Mace Head, on the west coast of Ireland. Mace Head regularly receives air from the midlatitude North Atlantic as well as from the UK and continental Europe, which allows them to estimate both the baseline and terrestrial emissions. Their emission estimates for the UK have been used to complement those reported to the UNFCCC for the period 1990-2007. Matross et al. (2006) derive regional-scale $CO_2$ flux estimates for summer 2004 in the northeast United States and southern Quebec using the STILT Lagrangian model in conjunction with aircraft and tower observations.

As mentioned in response to an earlier comment, we include a limited discussion of studies with regional inversions of other greenhouses gases, because they either use the same model or a similar procedure to establish the source-receptor relationship.

Lagrangian models are indeed not a prerequisite for a regional inversion, and we have edited the respective text as part of the changes made above.

3) Method sections (2-5)
The logical structure is often confusing. This regularly forces the authors to anticipate
for the next sections and thus make redundancies (e.g. the inversion technique is
introduced on p8 26-28 and at the beginning of sections 4 and 5.3). Presenting the
general frame of the inversion (5.3) could help solve for it.

The logical structure is to first introduce the ingredients of the inversion (data, priors, model=source-data relationship, baseline), then piece it together in a formal framework in section 5. We considered putting the general frame 5.3 of the inversion in front, as suggested by the referee, but this would mean we have to anticipate the detailed information about the ingredients that only follow later.

The mathematical formalism should be based on rigorous notations. Presently, the vector x and matrix T describe completely different things between section 3.2 and section 5.3. The spatial distribution of the fluxes within a region, which is implicitly contained in matrix T in section 5.3, and not in the vector of the prior estimate of the regional budgets x0, are called throughout the text "prior distributions".
The prior uncertainties derived from scientific publications and objective comparisons (section 4.1 and then extrapolated in section 5.3 at the regional scale) seem to be smaller than the artificial "uncertainty component of 50% of the seasonal amplitude" added in section 5.3 (even though it is difficult to understand what l15-17p16 really mean). Therefore, the derivation of the prior uncertainties seems artificial.
The introduction of the term with S in equation (2) raises questions. Why did not the authors rather introduce temporal correlations in the Co matrix to limit the changes in the weekly variations of the fluxes ?

We thank the referee for picking up the inconsistent notations. We have changed the wording regarding $x_0$ whenever a confusion regarding the within-region flux distribution and regionally integrated prior fluxes appeared likely. In the description of the transport matrix, we now distinguish between $T_g$, which links sources at the grid level to the observations, and T, which links the aggregated regional sources to the observations:

A transport matrix $T_g$ (unit s m$^{-1}$) is formed by dividing the dosage by the total emitted mass and multiplying by the area (m$^2$) of each surface grid cell. Each element of $T_g$ describes the atmospheric transport of a continuous emission of 1 g $CO_2$ m$^{-2}$ s$^{-1}$ from a given grid cell over the previous 4 days and subsequent contribution to the air concentration at the receptor (BHD or LAU) during each 1 h period. With x being a vector containing all grid cells and c a vector containing the concentration (unit g $CO_2$ m$^{-3}$) for all 1 h periods, this is written as $T_g$ x=c. Given $T_g$ and the measured concentrations c, the inversion developed in this work solves for the $CO_2$ fluxes x using a Bayesian optimisation, i.e., a statistical model that balances information from measurements with a priori knowledge about the fluxes (section 6).
Instead of solving on the grid scale, however, the fluxes in x are pre-aggregated into a set of regions (section 5.2) and a transport matrix T created by aggregating the grid cell in $T_g$ to reflect the regions in x,

$$Tx=c \qquad\qquad\qquad (1)$$

In addition, a priori flux maps are taken into account for the terrestrial and oceanic portions of the domain (section 4).

The prior uncertainties for the regionally integrated land fluxes are generally larger than could be assumed from the reported uncertainties at the grid scale. As described in section 5.3, full spatial correlation is assumed during aggregation, in order to reflect that within-region flux patterns are fixed and cannot be changed by the inversion. The resulting regional uncertainties are mostly larger than the additional 50% of the seasonal amplitude, unless the prior flux is near zero, i.e., some time in spring or autumn. The additional uncertainty component thus ensures that the inversion can also adjust those near-neutral fluxes, which would otherwise be practically prescribed and would introduce artificial seasonal turning points that cannot be altered by the inversion. For the majority of weeks, the prior flux uncertainty is largely formed by the aggregated grid-scale uncertainty.

The S term in equation (2) is an alternative to introducing temporal correlations in the prior covariance matrix. It has the advantage of being very transparent, i.e., by explicitly prescribing a smoothing scale of 5 kg CO2/m2/yr, we know that if posterior flux estimates exhibit an uncertainty as large as this, then they are basically only constrained by the smoothing term and not by the observations.

The logic behind the specific choice and configuration of the sensitivity tests on page 18 is not really convincing and does not seem to tackle some of the most critical sources

of errors in the inversion. What is the link between the modeling in NAME (l4p18) and the details given on (ii) later in page 18, which concern the spatial distribution of the fluxes within the regions ? Definitely, a practical assessment of the transport model uncertainties would have been interesting. I do not understand the test of sensitivity to the inclusion of the ocean fluxes in the modeling framework. What does l2p19 mean ? How to connect it to l30p28 ? In principle, the impact of the uncertainty in these fluxes should be correctly accounted for in the reference test. The point maybe be about "biases" but it is difficult to guess, in this section and when analyzing the results, whether the authors systematically make a rigorous use of the term "bias".

The description "(ii) the modeling in NAME" was indeed confusing, and we have edited the text to reflect it is about the within-region flux distribution:

These include (i) the $CO_2$ baseline, (ii) the flux distribution within each region, and (iii) the ocean prior fluxes.

Sensitivity case (iii) investigates how sensitive the inversion results are towards the ocean prior. The ocean prior is based on a low-resolution $pCO_2$ climatology and, while further away than the land regions immediately surrounding the stations, still influences the measurements (particularly at Baring Head, as seen in figure 7), so that we think it is worthwhile to isolate its impact. We have clarified how the ocean prior is "removed" in the text (also as part of our response to referee #1):

In a sensitivity test, we excluded the ocean prior to isolate its impact on the results. For this, the oceanic prior uncertainty was raised to $10^8$ Tg $CO_2$ yr$^{-1}$.

4) Result sections (7)
In general, the analysis are a too qualitative.
The robustness of the inversion needs to be better assessed through the analysis of the comparisons between prior and posterior CO2 vs. measurements. I feel that relying on the sensitivity tests to state that the results are robusts (l20-21p1, l8p29) is not really satisfying given that the sensitivity tests do not really sample the most critical sources of errors in the inversion system. The first subsection of section 7 could have been dedicated to such a detailed analysis of the CO2 concentration model - data misfits, providing an opportunity to exploit the long discussions on the station footprints to potentially correlate the highest misfits with specific transport conditions.

We have expanded the discussion of the model-data mismatch in section 7.4, also relating to main comment A) (we discuss the details there). The mismatch, especially at Lauder for the 15:00-16:00 LT time slot is now discussed in light of boundary layer dynamics as well as potential biases introduced by the lack of a diurnal cycle. We think that these components are the most likely causes behind the observed mismatch. Specific transport conditions could also have an impact, especially if there is a strong signal from Australia, but as discussed above, strong Australian signals are not very frequent whereas the observed mismatch in Figure 12 seems to be of a more continuous nature.

In order to ensure that the chi square statistics are right, the authors have to derive a 0.4ppm model error which is surprisingly low for 10magl stations surrounded by a complex topography, and which is strange given that the model data misfits often exceed the projection of the prior uncertainty in the obs space. As said in major comment B, I think that the authors made a wrong chi test but I also assume that this hardly explains such a low diagnostic of the model error. Figure 12 normalize the model data misfits by the prior uncertainty and the text hardly discusses absolute values (in ppm) of these misfits. These absolute values could reveal the skill of the model for simulating CO2 at the measurement sites. At least, figure 12 reveals large biases (and errors with a high temporal correlation over several months) whose consequences for the confidence in the inversion results should be better weighted.

As detailed in our response to an earlier comment, the chi-squared test had been computed correctly, but was incorrectly documented, which we corrected.

The 0.4 ppm refer to a minimum error applied before the chi test. Based on the chi test, a

global factor of 2.9 is then applied. This information was missing in the text and we have edited the respective paragraph in section 5.3:

Data uncertainty is calculated as the quadrature sum of the baseline uncertainty (Section **Fehler! Verweisquelle konnte nicht gefunden werden.**) and the $CO_2$ data uncertainty (Section 2). A minimum uncertainty of 0.4 ppm is assumed to account for uncertainties in the inverse modeling system as well as possible errors in the fossil fuel emission estimates. The final uncertainty is multiplied by 2.9, a value based on a goodness of fit analysis of the inverse model (reduced chi-squared statistic, as described below). The final data uncertainty is taken as the root mean square (quadrature) of both components. The square of the uncertainty populates the main diagonal of the data covariance matrix $C_d$. We assume no correlations between pairs of data points, so all off-diagonal elements of $C_d$ are set to zero.

We do not see the usefulness of presenting absolute values in Figure 12, as those can be adjusted heavily by the choice and balance of prior flux and data uncertainty. In the context of the chi test, the mismatch relative to the prior uncertainty is more meaningful.

The confidence in the inversion results at the weekly scale can also be weakened by the quite high week to week variations of the inverted weekly fluxes despite the term with the S matrix in equation 2. However, while the text takes time to discuss this penalty term in section 5.3, it skips the analysis of the week to week variations in the result sections. Does it lower the confidence in the monthly mean results or can it be explained by actual variations or through the variations of the observation constraint ?

It is true that the smoothing term could potentially weaken the confidence in the results. However, as discussed as part of our response to an earlier comment, our choice to use an explicit Gaussian smoother in the cost function over, e.g., temporal correlations, was based on its advantage in transparency. It will only considerably affect a particular flux estimate if the data or prior cannot constrain it to within 5 kg CO2/m2/yr, which is more than ten times larger than the largest prior flux from any grid cell. But it will nudge a flux estimate towards a smoother curve if there is little penalty, i.e., if two possible estimates come at roughly the same cost it will choose the smoother one. The posterior flux uncertainties are much smaller on a per-area scale than the value above, so we concluded that the smoother's effect is indeed limited to that nudging process.

One has to trust the authors that their inversion results can be directly compared to a sort of crude NIR total estimate in the abstract and in the introduction. However, the text in 7.3 reveals (even though it is often confusing) many differences between the type of fluxes covered by the inventory and the inversion, and that the inventory provide enough details to filter some major flux components that cannot be accounted for by the inversion, so that a more relevant budget could be derived for this comparison. Therefore, things should be presented differently (i.e. turned the other way around, starting with a presentation of the content of the NIR, and following with an extraction of a relevant budget from the NIR) and, in the abstract,"the sink [derived by the inversion]" should not be compared to "the reported 27 tGco2YR-1" (l. 26-27 p1) but to a more relevant combination of the NIR components.

We agree that the abstract could have been structured in the way suggested by the referee, by directly presenting a modified NIR estimate that we compare our estimate against. However, there is some merit in reporting the original NIR value, as it is more recognizable among the community. We would therefore like to keep the current structure where many flux components that could reconcile the two estimates are briefly presented in the abstract and then in much more detail in section 7.3.

5) Conclusion (8)
Given all my concerns that are detailed above, I think that the discussion misses the critical needs for the improvement of the inversion in the last paragraphs, and that it is too optimistic regarding the results in the first paragraph and at l1-3p31.

We have added to the last paragraph in the conclusions to point out that there are significant uncertainties, mainly with regard to annual mean flux estimates:

The inversion methodology developed here is a powerful tool to validate net regional $CO_2$ sinks in the New Zealand national inventory report. It offers an independent, top-down view on the national carbon budget. The limited sensitivity to the northern half of the North Island, as well as baseline errors, can lead to large uncertainties for annual mean flux estimates in some regions. Improving on these factors in future studies can further increase the usefulness of the top-down approach.

- some questions

* why do not the author assimilate 14:00-15:00 data ? in order to save computations ?

This is one reason, yes. Another reason is that the footprints for the 13:00-14:00 and 15:00-16:00 periods are highly correlated and visually almost indistinguishable from each other. It is likely that the 14:00-15:00 footprints would be, too. Therefore adding them would amount to essentially blowing up the transport matrix with equations that are not linearly independent and add little information. This would be similar to just averaging over all three data periods.

* p6 l3 vs l7: it seems that this site should be strongly influenced by the emissions of Wellington. Can the authors provide more details (e.g. statistics) on this topic than at l23-24p20 ?

The influence is mostly reflected by the NZ fossil fuel contribution to the observed anomalies at BHD in Figure 7 (red bars). It is comparable in size to the ocean contribution. Contributions from other fossil hotspots, like Auckland, are seen at both stations and are quite small at most times as seen in the lower panel for LAU.

* l31p11, l12p2, l12-16p24: if focusing on fossil fuel emissions, what is the difference between EDGAR and NIR ? can we assume that NIR has a far more precise estimate than EDGAR ?

We think it is likely that the NIR has a better estimate for fossil emissions in New Zealand. Originally, we intended to use their values, but it turned out that there are no mapped distributions of emissions available for the public. While the two estimates are slightly different at the national scale, we discuss this difference and its implications in section 7.3.

- Sample of minor issues illustrating some of the general comments above

* the text often forgets to be precise about the fluxes that are discussed (l13p1-> natural, l.12 p2: CO2 emissions -> anthropogenic accounting for land use or fossil fuel only ?; first sentence and line 13 of abstract and 1st paragraph page 4: precise that you target natural fluxes; l27p8: the system solves for natural fluxes; l2-3p9 sinks and sources: natural sinks and sources...); on the same topic, lines 12-14 p24: as it is, these sentences do not make sense since land use change emissions are not fossil fuel emissions.

The inversion solves for net $CO_2$ fluxes that contain anthropogenic and natural elements. This was already pointed out in the 2nd paragraph in section 5.3, but we have made it more clear by adding the word "net" in the abstract and the first sentence of section 5.3:

(abstract) This approach infers net air-sea and air-land $CO_2$ fluxes from measurement records, using back-trajectory simulations from the Numerical Atmospheric dispersion Modeling Environment (NAME) Lagrangian dispersion model, driven by meteorology from the New Zealand Limited Area Model (NZLAM) weather prediction model.

(section 5.3) The aim of the inverse method is to estimate a net $CO_2$ flux from every region and for every week between 2011 and 2013 using a Bayesian approach (Gurney et al., 2004; Tarantola, 2005; Steinkamp and Gruber, 2013).

* discussing RBM in section 2 and 5.2 is a bit strange but the authors are embarrassed

with the fact that the partitioning of the fluxes accounts for the future inclusion of RBM data in the system for future experiments.

As explained in the manuscript, the RBM station was included in the design process for the inversion, to allow for direct comparisons between our results here and the results of future papers. Our mentioning of RBM in section 5.2 offers maximum transparency and insight of what we did.

* examples of abusive shortcuts: l13 p1 (from measurement records), l16 p2 (locally present vegetation), p822-24, l7p17 (which quantity does this number corresponds to ? C is a cov matrix). . .

In the abstract and introduction we give a summary and brief introduction into what we studied, while all necessary details follow in their respective sections. We do not think repeating these details adds to the value of the manuscript.

We edited the text at l7p17 to clarify the presented number is squared before populating the covariance matrix:

The diagonal matrix $C_s$ contains the squares of values representing the strength of the smoother.

* examples of awkward and confusing sentences: l2-3p3, l10-11p3, all sentences from l14 to l21p3, l6p4, l23-24p9 (there is a long discussion on the biomes and land use maps on page 9 and 10 but we hardly understand how they will be used), the first paragraph of 5.1, "monthly standard deviations" l20p13, l4p15,l15-17p16, l20-21p16, l4-6p17, l10p17,l14-15p18, the whole paragraph (ii) on p18, l21p21,12-14 p24, l20p24, l21-23p26, l22-25p27, l26-28p27.. . .

Some of these examples refer to earlier comments and we are confident that we were able to clarify the text as part of our responses there.

The remark about the first paragraph of 5.1 seems to relate to the fourth paragraph of 5.1. There, we inform the reader briefly about what we did, but refer to appendix B for more details. As such, we do not think repeating those details at that point would be helpful.

* l18p5: the link between the PBL and the horizontal extent of the footprint of the measurements is a bit confusing

We have added a reference to section 7.4 where this link is discussed in more detail.

* the logic behind the model representativeness error computation is not obvious (p5l28-32), in principle, the STD of the concentration variability at the 5-min scale does not correspond to the skill of the model to represent hourly measurement averages; the assumption underlying this computation should be explained

We have edited the text to avoid the reader getting the impression we are trying to explicitly link the skill of the model to represent hourly measurement averages with the data variability at the 5-min scale:

This uncertainty is generally much greater than the measurement imprecision, as it reflects real atmospheric variability, and is instead intended to capture representativeness errors such as the measurement failing to represent the mean of a model box or the model failing to represent the specific conditions at an individual location.

 As part of our response to an earlier comment we have also added additional information on data uncertainty in section 5.3.

* p8 l8-11: the normalization (if I understand it correctly) does not make sense and just

loses information.

The normalization simplifies spatial integration and computation of regional responses. It does not lose information. The only information "lost" is the integrated value over the whole domain, but this value is a direct result from the amount of $CO_2$ released in the NAME model and can be chosen freely without any effect on the inversion (see also section 3.2).

* the text often forgets to associate numbers with a time or space scale (e.g. l31-32p10, 3rd paragraph of page 11, l7p24...)

We have clarified in the text in l31-32p10:
An uncertainty estimate is computed for the *a priori* $CO_2$ flux from each grid cell. Based on Keller et al. (2014) and personal communication with the authors, we assign 10% of the flux as uncertainty for pasture land.

We were unable to localize the number the referee is referring to in the 3rd paragraph of page 11.

We have clarified in the text in l7p24:
The NIR estimates do not come with an overall uncertainty, but based on their reporting of typical uncertainty for individual ecosystems, and personal communication, an approximate figure of 50% of the flux value was identified.

* section 6 is poorly connected to the other sections, and the part of section 6 before the start of 6.1 sometimes sounds like a summary of the subsections 6.1 and 6.2

This has been answered as part of an earlier comment.

*l14-17p14 and l25-29p19: the diagnostic assesses the sensitivity to the fossil fuel emissions in the part of Australia that is in the modeling domain, not the sensitivity to all fluxes in the whole Australia; in other places, and especially in the conclusion, the paper will state that potential errors from Australia need to be handled.

We have clarified this:
This is based on an analysis suggesting generally low sensitivity of $CO_2$ measurements at our stations in New Zealand to fluxes from the Australian region…

* the logic behind the justification of the partition of the ocean regions in the second half of p14 is not really clear.

The partitioning of the ocean regions is guided by objective measures where available, but contains parameters that had to be chosen subjectively, as laid out in section 5.2.

* l5p16: it is not the "uncertainty in the inverse modeling system" (which is something that would be difficult to quantify), it is the likely the uncertainty in the transport model

We have corrected this as part of our response to an earlier comment.

* l7-10 p23 are poorly connected to the analysis above

This sentence provides additional information in the context of regional drought, which we believe is useful.

* l17-20p27 are wrong, if posterior uncertainties are low, negative correlations between these uncertainties do not mean that the inversion is unable to distinguish the corresponding fluxes; in such cases, it would just mean that the residual uncertainty in the corresponding fluxes is due to such a problem of separation

Mathematically, any negative correlations mean the sum of the fluxes is better constrained than it would be if the correlations were zero. However, our choice of wording ("being unable to distinguish") seemed too strong and led to confusion, so we edited the text:

Strong negative correlations between two regions would indicate that the inversion has difficulties to distinguish their individual flux components with the available data, while their sum is better constrained. Similarly, positive correlations are indicative of the difference of flux components being constrained better than each individually.

\* labels in the figures are often difficult to read

We have slightly increased the labels in Figure 8, where they appeared to be a little smaller than in the other figures. We are happy to respond to specific comments about which labels to change, but also think that the present sizes are readable provided figures will not be sized down.

\*l22-24 p21: at this stage of the manuscript, we do not know how the additional uncertainty from the sensitivity tests is included in the figure (this will be explained at l21-25p28 which are quite confusing)

We have added a reference to section 7.4 in the caption of Figure 10 to clarify:

The cyan shade represents the extra uncertainty obtained from the sensitivity cases (section 7.4).

---

## Author Response (AR2)

**Revision of ACP manuscript acp-2016-254**

**"*Atmospheric CO$_2$ observations and models suggest strong carbon uptake by forests in New Zealand*"**

by K. Steinkamp, S.E. Mikaloff Fletcher, G. Brailsford, D. Smale, S. Moore, E.D. Keller, W.T. Baisden, H. Mukai and B.B. Stephens

1 December 2016

Dear Editors,

The co-authors and I would like to thank the anonymous referees #1 and #2 for providing constructive reviews. They helped us to improve and clarify the paper. We have responded fully to both reviews below, and we would be delighted to submit a revised manuscript.

We thank you for your continued consideration of this manuscript.

Sincerely,

Kay Steinkamp

**Anonymous Referee #1 – Interactive Comment**

*Although the involved changes in the manuscript are sometimes minimal, my comments have been in most cases sufficiently well responded, and I recommend publication after a very small number of further minor changes (see below). I have read with great interest the comments of referee 2, as my own expertise is more limited, but won't comment on the related improvements in the manuscript. I note however that the new figure 14 is interesting and seems reassuring about the eventual systematic errors caused by the diurnal course; but the weakness and apparent randomness of the effect is somewhat surprising.*

We thank the reviewer for his or her constructive comments. When we set up the inversion, we created a very small region around each site to minimise the impact of local variability at the sites on the larger scale flux estimates. Much of the bias created by the diurnal cycle in our synthetic data experiment is captured by our Lauder local area region, as discussed in the paper and in our response to reviewer #2 below.

*There is one thing I would wish to be improved: in Table 1 (originally on page 44) an unclear division of both Islands into two regions is used; in their answer to this comment, the authors state that the definition of the division can be found by using Figure 10 etc.; but, first, the division of the*

*North Island is not described there; and, second, the South Island is apparently divided into two regions which overlap each other, which is odd but which is not further explained.*

Some of the region labels in Figure 10 were indeed wrong, in particular such that the South Island West and East regions seemingly overlapped. We have corrected the labels to correctly reflect the division. Thank you for catching this.

*A second point: Concerning my earlier comment on page 19 lines 4-29: The change proposed by the authors is good, but it has not been executed in the new version (see page 20 line 18-22 in that version).*

That change was indeed not executed – we apologize and have executed it in the revised version

**Anonymous Referee #2 – Interactive Comment**

*Steinkamp et al. have applied major changes to their manuscript "Atmospheric CO2 observations and models suggest strong carbon uptake by forests in New Zealand". I am not entirely satisfied with the answers they gave to my comments but I acknowledge that given the interesting material brought by this study, some of these comments could sound like a bit too meticulous. My main remaining issue is that I feel that the text (including the new and revised parts) is still often confusing and it often lacks of structure and rigor. I also feel that the authors sometimes provide weak answers in order to avoid accounting for my comments and to avoid new analysis and discussions (in particular new analysis of the modeled vs measured concentration timeseries). Finally, a significant number of my comments (either comprised within my major comments or in my list of minor comments) were misunderstood. In conclusion I would still push for a major and general improvement of the text but the scientific issues I would still raise are minor.*

We thank the reviewer for this feedback.  We regret that the reviewer feels we have attempted to avoid new analysis or discussion in response to his or her comments.  This was not our intention.  For example, the addition of an OSSE represented a major piece of new work for the paper, which we felt was appropriate given the reviewers serious concerns about bias from diurnal cycles.

In response to Referee #2's ongoing concerns about the quality of the text, we have asked a colleague who is not a co-author on the paper to review our manuscript with fresh eyes and provide suggestions about where the text could be improved.  His suggestions have been implemented in the revised manuscript, in the absence of further feedback from the reviewer.

*Here is a list of specific problems I see in their answers and in the*

*corresponding corrections to the manuscript:*

*- Regarding the test of sensitivity to the NEE diurnal cycle: the most straightforward test we could have expected was a new experiment using the real data and the reference inversion set-up, but the prior estimate of the NEE with diurnal variations. Comparing the reference inversion to this one would have provided a clear characterization of the impact of this diurnal cycle in the reference inversion results, and an evaluation of the confidence in these results. I do not say that the authors had to make this specific experiment, but their OSSE and of their analysis to make a demonstration of the weak impact of the diurnal cycle can sometimes be a bit puzzling:*

The reviewer seems to misunderstand our inverse modelling framework. We minimise differences between weekly prior fluxes and posterior fluxes in the cost function (Equation 2). Since the a priori flux constraint is applied on a weekly basis, adding diurnal variability to the prior would have no impact on the results of our inversion, unless the weekly mean flux were also changed.

While the reviewer did not request we undertake an observing system simulation experiment (OSSE), we felt this was the most comprehensive way to evaluate potential biases introduced by neglecting diurnal variability.

*\* I am not 100% sure about the way they derived the diurnal cycle of the synthetic NEE: "the diurnal variation in GPP is based on hourly solar insolation": does it mean that it is exactly set-up based on the relative diurnal variations of the solar insolation ? I cannot see numbers illustrating the amplitude of this cycle (so it is difficult to check that it is "exaggerates" actual diurnal variations).*

The reviewer is correct that in our experiment the GPP varies exactly according to the diurnal variation in solar insolation. This exaggerates the diurnal variation in NEE fluxes because, in reality, photosynthesis rates reduce at midday and respiration rates increase with temperature so that the peak amplitude is lower than would be implied by the effect of solar insolation alone. The amplitude of our simulated diurnal cycle is 1331 TgC yr$^{-1}$ in summer (Dec-Feb average) and 690 TgC yr$^{-1}$ in winter (June-Aug average) integrated across all New Zealand land regions.  We also show it in figure 1 below, but have not added this figure to the manuscript in order to avoid excessive length.

[Figure]

**Figure 1.** Diurnal cycle of $CO_2$ fluxes during summer (blue, Dec-Feb average) and winter (red, Jun-Aug average) integrated across all New Zealand, estimated for our synthetic data experiment. Uptake is negative.

In order to clarify this point, we have changed the text on p.30 lines 9-10 as follows.

*"The diurnal variation in GPP is directly proportional to the relative amount of hourly solar insolation, and HR is assumed to occur at a constant rate spread evenly throughout the day. There are a number of aspects of plant physiology and ecosystem biogeochemistry that cause actual diurnal variation in NEE to be more muted than the solar-radiation driven pattern we have modelled. These include reductions in photosynthesis during the middle of the day and afternoon as stomata close due to drought or leaf temperature stress. Similarly, reduced respiration can be expected at night due to cooler temperatures. "*

*\* the authors connect this OSSE to the discussion on the model-data mismatch at LAU for 15:00-16:00, but in order to assess whether uncertainties in the NEE diurnal cycle impacts these mismatch, they could had looked at the 15:00-16:00 concentrations modeled when using the flat NEE vs. the one with diurnal variations (instead of trying to read such a comparison out of the inversion results). As a consequence, they assimilate 15:00-16:00 data in the OSSE instead of both 13:00-14:00 and 15:00-16:00 as in the reference experiment. This and having several other parameters that are different in the OSSE compared to the reference inversion does not help evaluate the robustness of the reference inversion with regard to uncertainties in the NEE diurnal cycle.*

There are two ways to look at the impacts of the diurnal cycle in the framework of our synthetic data experiment: 1) compare the modelled mole fractions with and without the diurnal cycle (data space); 2) compare the posterior fluxes with and without the diurnal cycle (flux space).

We chose to undertake the analysis in flux space, because it allows us to quantify potential biases in our flux estimates, which are the central results presented in the paper. We appreciate the referee's suggestion that we undertake this comparison in data space instead, but we feel the flux comparison is more relevant to our final result. In order to satisfy the reviewer's request without making the paper excessively long, we present analysis in data space (below) but omit it from the revised paper.

In Figure 2, we show the difference that the diurnal cycle makes to simulated atmospheric $CO_2$ at Baring Head and Lauder. At Baring Head, the bias is small compared to the variance, and there is some seasonal signal, with lower $XCO_2$ in (austral) summer, when the diurnal cycle is resolved. In flux space, this should translate into 1) little overall bias in regional annual mean fluxes and 2) slightly increased $CO_2$ uptake in summer. Both of these conclusions have also been reached at in our discussion in the manuscript. Furthermore, by doing the analysis in flux space, we could go into a bit more detail, such as pinpointing the regions in which such a seasonal impact is recognizable.

At Lauder, we can see a clear positive bias along with a very small seasonal signal. The fact that the bias is positive and not negative, as one would expect when including diurnal flux variations (at least in summer), is discussed in the manuscript as well. That bias translated to a positive flux anomaly in flux space. As seen from Fig. 14 in the manuscript, that positive anomaly is mostly contained in the local region around Lauder (region 14). Capturing such local biases was part of our motivation for including the local regions around measurement stations.

To summarize, we believe undertaking the analysis in flux space provides more detail and a more direct link to the results of the reference inversion, while also capturing the information obtainable from a comparison in data space.

As for the second part of the comment, we did not include a comparison for 13:00-14:00 data due to the computational cost associated with the new, hourly footprints on top of an already substantial piece of additional work. We chose to use the 15:00-16:00 time slot for our analysis, as it is closer to the observed minimum $XCO_2$ in the afternoon and should be affected more strongly by the lack of diurnal flux variations in our reference model.

We also would like to emphasize that for our synthetic experiment, we kept as many parameters as possible the same as in the reference case. However, it was unavoidable to make some changes, e.g. the new dataset of hourly fluxes does not have the same weekly averages as the prior

used in the reference inversion, so we had to use the new flux map in both synthetic runs for consistency.

[Figure]

Figure 2. Data mismatch for synthetic experiment with and without a diurnal flux cycle. Solid lines are Loess fits with a 3 month window.

*the analysis added in 7.4 to comment on the impact of uncertainties in the diurnal cycle based on Figure 14 are often confusing but yes, Figure 14 seem to indicate that this impact is small at the annual scale. The much larger impact on results at the monthly scale and for the seasonal cycle could have been more emphasized (while the claim by the author that the truth and the inversion agrees within their uncertainty is misleading). At least, for regions 7, 12, 14 and 15 it is clearly significant and often larger than the other sources of uncertainties accounted for in the estimate of the posterior uncertainty, which shows that this uncertainty should be increased to account for this additional source of error at the monthly scale.*

We discussed the higher weekly variability in regions 12, 14, and 15 in paragraph 3 of the Diurnal Variability subsection of 7.4 in the previous submission. However, In response to the reviewer's concerns, we have revised the text of section 7.4 as follows. New text is shown in *itallics*.

Unrepresented diurnal variability led to biases in the annual mean flux estimates for some regions in our inversion, but these errors were much smaller than our uncertainty estimates for most regions *on an annual scale* (Figure 14)...

*On a weekly time scale, estimates generally agree to within their uncertainties for most regions, with the exception of the Eastern South Island (Regions 12 and 15), the southern Central North Island (region 7), and the local area region around Lauder (region 14.)* Regions 12 and 15 (eastern and south-eastern South Island) *show* an increased sink late in the year (as part of the 2012/2013 summer) but a smaller sink early in the year (as part of the 2011/2012 summer), *suggesting that there may be a seasonal bias in our inverse methodology for the eastern South Island. Likewise, diurnal cycle bias leads to significant weekly errors in the central North Island (region 7), although with less seasonal coherence. The Lauder local region (region 14) was created to capture local signals that are not well represented in our inverse model, including diurnal variability, and prevent them from biasing the inverse estimates on larger spatial scales. Thus the larger errors for this region are expected.*

*- Regarding the need for a general improvement of the text: as I explicitly said, I only listed examples illustrating such a need in my previous review, but I did not conduct an exhaustive listing of the problems. However, the authors "sought to improve the quality of the text in places specifically pointed out by the referee" when they did not just rebut the corresponding concerns. As a consequence, they did not conduct the detailed proofreading which was needed, and which is thus still needed.*

We regret that the reviewer feels the revised manuscript was poorly written, as we have made an earnest attempt to put forward a high quality manuscript. We have asked a colleague who is not a co-author on the paper, Dr. Hinrich Schaefer, to provide an independent review of the writing for clarity, rigor, and structure. He has pointed out a number of places where the text could be improved or clarified, and we have implemented these changes throughout the revised manuscript, but he did not recommend any major structural changes to the text. We hope our revised manuscript addresses the reviewer's concerns on this point.

*- I am still a bit puzzled regarding the topic of the model error. I am really confused by the authors' explanations regarding the precise computation of the model error in page 16-17 and how it can be consistent with what is said in page 18 regarding the 0.4ppm value. The text does not give the typical value for the model error arising from such computations. Furthermore, the authors refused to conduct some more detailed analysis*

*of the prior and posterior model vs measured concentration timeseries (which would have been useful to discuss the theoretical value that they derive for the model error), and more generally to conduct some additional evaluation of the transport model, assuming that "a more in depth analysis of transport model bias is outside the scope of this paper".*

We agree that the phrasing on page 18 can be improved. As detailed on page 16-17, the 0.4 ppm refer to the minimum uncertainty, and hence are only part of the final uncertainty. The latter is calculated as described further on page 16-17.
We have clarified this in the text on page 18:

Computing the data uncertainty as described above ensures $X^2 \approx 1$, which means (…)

In addition, we have added a sentence on page 16-17 to provide an average value and a range of the final uncertainties:

The resulting uncertainty is taken as the root mean square (quadrature) of both components and has a minimum value of 1.16 ppm. The mean uncertainty is 1.91 ppm and 95% of the values are within the 1.16 to 4.56 ppm range.

*-My comment on the weakness of the configuration of the sensitivity test has not really been accounted for even though these tests are used to provide an assessment of the robustness of the inversion results*

We respectfully disagree.  This point was answered in our original revision and response letter by addressing specific questions the reviewer raised in the response letter and expanding discussion of errors due to transport model uncertainty and unaccounted for diurnal variability in the first revised manuscript.

*- Some technical points:*
*\* the mathematical notations are still problematic, e.g. Tg and T are multiplied by the same vector on page 9 or the dimensions of x and T change from page 9 to 17*

In order to prevent potential confusion regarding equations defined on the model grid with those defined on aggregated regions, we now distinguish between fluxes $x_g$ (on the grid scale) and $x$ (aggregated regions). The respective text on page 9 now reads:

With $x_g$ being a vector containing all grid cells and $c$ a vector containing the concentration (unit g $CO_2$ $m^{-3}$) for all 1 h periods, this is written as $T_g x_g = c$. Given $T_g$ and the measured concentrations $c$, the aim is to solve for the $CO_2$ fluxes $x_g$ using a Bayesian inversion, i.e., a statistical model that balances information from measurements with *a priori* knowledge about the fluxes (section 6).

Instead of solving on the grid scale, the fluxes in $x_g$ are pre-aggregated into a set of regional fluxes (section 5.2), $x$, and a transport matrix T is created by aggregating grid cells in $T_g$ to reflect the regions in $x$,

$$Tx = c \tag{1}$$

The Bayesian inversion developed here solves for $x$. In addition, *a priori* flux maps are taken into account for the terrestrial and oceanic portions of the domain (section 4).

*I do not understand the justification for the S term in equation (2) based on a matter of "transparency".*

As we described as part of our response to the original comment about the S term, we think the ability to quantify the strength of the smoother with respect to the other terms in the cost function is a transparent way of introducing this additional constraint. We do not say that imposing temporal correlations instead cannot be transparent, but merely that formulating the smoothing constraint as an explicit term in the cost function makes a transparent interpretation of its impact straightforward.

*I think that the authors misunderstood my point regarding their estimate of the representativeness error. I just think that strictly speaking, the 5-min scale variability of the measurements does not reflect the spatial variability of the hourly mean measurements in model box.*

We agree on this. We did not intent to suggest the spatial variability in the model can be represented by the 5-min data variability. Rather, this 5-min variability can be used to estimate error related to differing temporal resolutions of model (1 hour) and data (5 minutes). We have rewritten the respective sentences on page 6:

For both stations, one standard deviation of a 5-minute measurement interval is taken as random data uncertainty for the hourly mean. This uncertainty is generally much greater than the measurement precision, as it reflects real atmospheric variability, and is instead intended to capture representativeness errors such as different temporal resolutions of model and data or the model failing to represent the specific conditions at an individual location.

* Regarding the meaning of the correlations between posterior uncertainties in two flux components: the author insist in writing something wrong i.e. "strong negative correlations between two regions would indicate that the inversion has difficulties to distinguish their individual flux components with the available data". Again, if the uncertainty reduction for the flux in each region is about 99% but the remaining (posterior) insignificant uncertainties in each region have a -0.9 correlation, we would still have to say that the inversion managed to distinguish their individual flux components.

We have difficulties following the reviewer's logic here; suppose a posterior correlation matrix for two flux components has (i) a -0.9 off-diagonal value, (ii) a -0.1 off-diagonal value. No matter what the variances are, the sum of both flux components will have a lower uncertainty in case (i) than in case (ii). We agree that with stronger uncertainty reduction the significance of the correlation diminishes in the context of distinguishability of the flux components, which is why we weakened our statement in response to the original comment. However, it is still a valid statement. The reviewer seems to suggest there is some kind of threshold, i.e. that once the uncertainty reduction is large enough (the reviewer mentions 99%) the distinguishability issue disappears completely, which is not true. In any case, in the context of our study, typical uncertainty reductions are in the range of 30% to 60%, except for the local regions around the sites where it can reach 80%, arguably not high enough to justify the claim that the inversion can resolve all regions perfectly. One reason we added that statement in the manuscript was to make it clear that we do not claim our inversion to perfectly distinguish all flux components.

*\* in general: the authors easily use the term "bias" when discussing random or varying errors*

We have examined each use of the term bias in the text, and replaced it with 'error' where appropriate.